# Quenched random mass disorder in the large N theory of vector bosons

Han Ma

*Perimeter Institute for Theoretical Physics,*
*Waterloo, Ontario N2L 2Y5, Canada*

## Abstract

We study the critical bosonic O(N) vector model with quenched random mass disorder in the large N limit. Due to the replicated action which is sometimes not bounded from below, we avoid the replica trick and adopt a traditional approach to directly compute the disorder averaged physical observables. At $N = \infty$, we can exactly solve the disordered model. The resulting low energy behavior can be described by two scale invariant theories, one of which has an intrinsic scale. At finite $N$, we find that the previously proposed attractive disordered fixed point at $d = 2$ continues to exist at $d = 2 + \epsilon$ spatial dimensions. We also studied the system in the $3 < d < 4$ spatial dimensions where the disorder is relevant at the Gaussian fixed point. However, no physical attractive fixed point is found right below four spatial dimensions. Nevertheless, the stable fixed point at $2 + \epsilon$ dimensions can still survive at $d = 3$ where the system has experimental realizations. Some critical exponents are predicted in order to be checked by future numerics and experiments.

**CONTENTS**

## I.   INTRODUCTION

Systems with quenched disorder are of great interest as they are expected to host novel universal behaviors. In the system of repulsively interacting bosons, even weak disorder would drive the superfluid-insulator transition to a new universality class, which describes a superfluid-glass transition[1–6]. This has attracted attention for decades due to its experimental relevance[7–13]. Meanwhile, it remains a challenging theoretical problem because there is still a lack of field theoretical understanding in the long distance limit, especially for systems in higher spatial dimensions.

Such a bosonic problem can be formulated as a $d+1$ dimensional [1] effective Euclidean quantum field theory with random couplings. In general, there can be random fields, random mass, random chemical potential and random interaction, each of which affects the system in a different way. Particularly, the bosonic critical theories with quenched mass or potential disorder have been studied extensively. Many works are done for $d = 1$ systems using exact Bosonization techniques[1, 14–16]. Higher dimensional systems have been studied numerically[17–23]. Theoretically, besides some general developments [24–26], perturbative methods are still largely used to shed light on the weak disorder problems.

We study a theory of N-component vector bosons. For each component being a real field, the system has O(N) symmetry. It is well-known that the clean O(N) vector model has a continuous phase transition at the famous Wilson-Fisher fixed point below three dimensions [27–29]. The limit of $N \to \infty$ gives a critical theory of generalized free fields. Such large N theories coupled to a random field are comprehensively discussed[30]. Here, we are particularly interested in mass disorder. As this quenched disorder is a (marginally) relevant perturbation at $d \geq 2$, around the fixed point at $N = \infty$ and $d = 2$, we use the double expansion of $1/N$ and $\epsilon = d - 2$ to access a system with finite N and higher spatial dimensions, hopefully $\epsilon = 1$. Previous works gave the RG flow around this UV fixed point[4] and suggested that at $d = 2$ there is an attractive disordered fixed point in the IR[5, 6].

In this work, we verify that this stable fixed point continues to exist in $d = 2 + \epsilon$ spatial dimensions, as schematically shown in Fig. 1(a). If we go to higher dimensions, the fixed point structure evolves in the following ways, as presented in Fig. 1. Firstly, as the spatial dimension exceeds $d = 3$, the Wilson-Fisher fixed point moves into the lower half plane and becomes unstable along both directions while the Gaussion fixed point becomes stable against the interaction. Secondly, the stable disordered fixed point continuously moves towards the strong disorder regime. It is

---

[1] Hereinafter, $d$ stands for spatial dimension and we fix the temporal dimension to be one.

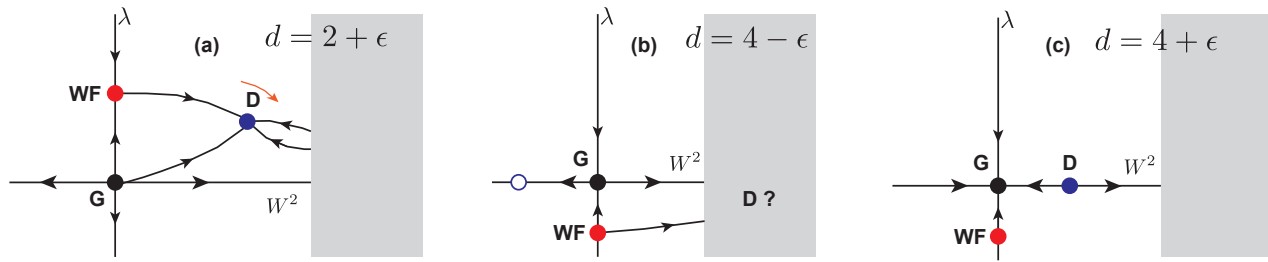

FIG. 1. Schematic fixed point structure at $d$ spatial dimensions is presented as our main result of this work. $\lambda$ denotes the interaction and $W^2$ quantifies the randomness. They span the coupling space of interest. Gray zones represent the non-perturbative regions which cannot be reached in our study. (a) At $d = 2 + \epsilon$, there is a IR stable fixed point (blue dot labeled by "D") at finite randomness and finite interaction, together with the Gaussian fixed point (black dot labeled by "G") and Wilson-Fisher fixed point (red dot labeled by "WF"). As we increase the spatial dimension, the disordered fixed point moves rightward as shown by the orange arrow. (b) At $d = 4 - \epsilon$, the theory flows to the non-perturbative region. Meanwhile, there exists a stable but unphysical fixed point (blue circle) at negative randomness. (c) Above four spatial dimensions, the originally unphysical fixed point enters the right half plane (blue dot labeled by "D"). Its stability is changed. As a result, the Gaussian fixed point becomes stable under weak disorder.

expected to either enter the non-perturbative regime or annihilate with another unstable fixed point at critical spatial dimension $d_c$, meaning we can no longer keep track of it. At $d = 4 - \epsilon$, our calculation finds no physical fixed point at finite randomness for free bosons (Fig.1(b)). This is also suggested by the unstable replicated action of the random mass Gaussian theory. However, this perturbative study is failed to predict the RG flow in the strong disorder regime. It is possible that the interaction becomes relevant at large randomness even above three spatial dimensions and hence stabilizes the theory. Thirdly, at $d = 4 - \epsilon$, an unphysical fixed point appears at negative randomness as shown in Fig. 1(b). It approaches the Gaussian fixed point as the spatial dimension goes up. As $d$ becomes larger than four, it enters the right half plane and is no longer stable under disorder. As a result, the Gaussian fixed point becomes stable (Fig. 1(c)). It is worth mentioning that in the deep IR, the disorder distribution is unlikely to remain Gaussian, as implied in Sec. VI B. So, besides the mean and variance, we also need the higher moments of the distribution to characterize the disordered fixed point. This is beyond our scope and requires non-perturbative methods.

Previous works mentioned above make use of the replica trick. This is a standard way of doing disorder averaging. It is done by considering $n$ copies of the same system and integrating over the disordered coupling with respect to its probability measure in the total partition function. This

produces an effective replicated action. One can further study its RG flow and compute disorder averaged observables. However, the replicated action is not always well-defined. For example, the O(N) model at $d > 3$, which will be discussed in the later sections, does not have a replicated action which is bounded from below. Thus, instead of using the replica method, we adopt a traditional approach which is used to study the problem of random impurity scattering[31–33]. In this way, we compute the correlation functions as a series expansion of the random coupling, followed by a truncation at finite order. Then, we can do the disorder averaging order by order. This would be a good approximation as we suppress the random coupling by a factor of $\frac{1}{\sqrt{N}}$ throughout the paper. Sec. II B reviews its diagrammatic representation in detail. Then, based on the general functional formalism in Sec. II, we study the critical O(N) vector model with Gaussian random mass disorder in $2 < d < 4$ spatial dimensions in Sec. III $\sim$ Sec. VI. It is known that the clean critical point is at the Wilson-Fisher fixed point when $d < 3$ and at the Gaussian fixed point when $d > 3$. Based on the criterion of the relevant disorder given in Sec. III, the disorder is found to be relevant at the Wilson-Fisher fixed point for $2 < d < 3$, while the disorder is relevant at the Gaussian fixed point when $3 < d < 4$. We proceed to study the $N = \infty$ system in Sec. IV, where the system is described by a theory of generalized free fields. At $3 < d < 4$, the disorder is simply decoupled from the Gaussian theory. At $2 < d < 3$, the system flows to a new fixed point where all the anomalous dimensions of the fundamental and composite operators can be obtained by exactly computing the correlation functions and then taking the low energy limit. These results have an alternative understanding if we introduce an intrinsic scale to the system. Around the $d = 2$ and $d = 4$ fixed points at $N = \infty$, we use the $\epsilon = d - 2$ and $\epsilon = 4 - d$ expansions, respectively in Sec. V and Sec. VI, together with $1/N$ expansion to study the system at finite N and small $\epsilon$. Furthermore, the $1/N^2$ correction at $d = 4 - \epsilon$ is given in App. G. Finally, Sec. VII summarizes the results and gives a discussion on the fixed point structure in $2 < d < 4$. Appendices A $\sim$ G contains more detailed calculations.

## II. GENERAL FORMALISM

In general, we consider a $d + 1$ dimensional scale invariant theory, such as a conformal field theory, perturbed by the quenched disorder. Particularly, we deform a clean theory $S_0$ by a local scaling operator $\mathcal{O}_{x,\tau}$ whose random coupling $J_x$ is only a function of space coordinate. In Euclidean spacetime, the deformed partition function is

$$Z[J] = \int \mathcal{D}\phi \; e^{-S_0[\phi] - \int d^d x d\tau J_x \mathcal{O}_{x,\tau}[\phi]}. \tag{2.1}$$

Both $S_0$ and $\mathcal{O}_{x,\tau}$ are functionals of fundamental fields $\phi_{x,\tau}$. The disorder $J_x$ is quenched, meaning that it is infinitely long-range correlated along the time direction. It is drawn from a distribution

$P[J]$ which is fully determined by its moments $\overline{J_{x_1} \ldots J_{x_\ell}} = \int \mathcal{D}J \; P[J] \; J_{x_1} \ldots J_{x_\ell}$ or cumulants. We are especially interested in the Gaussian distribution where $\overline{J_x} = 0$, $\overline{J_x J_{x'}} = W^2 \delta^d(x - x')$ and all higher cumulants of $J_x$ are zero. Equivalently, the distribution is

$$P[J] = \frac{1}{\mathcal{N}} \exp\left\{ -\frac{1}{2W^2} \int d^d x \; J_x^2 \right\}, \tag{2.2}$$

where $\mathcal{N} = (2\pi W^2)^{V/2}$ is the normalization factor such that $\int \mathcal{D}J \; P[J] = 1$ and $V$ is the number of sites in the system. $W^2$ is the variance characterizing the width of the Gaussian distribution, which also quantifies the randomness of the coupling. As a result, the partition function in Eq. (2.1), as well as other observables, are functions of the disorder realization. Ultimately, the quantities of physical interests need disorder averaging.

In the conventional functional formulation, the correlation functions of a particular operator $\mathbf{O}_{x,\tau}$, including $\mathcal{O}_{x,\tau}$ and its composite operators, such as $\mathcal{O}_{x,\tau}^2$, $\mathcal{O}_{x,\tau}^3$, etc., are encoded in the generating functional. For example, a m-point connected correlation function is given by

$$\langle \mathbf{O}_{x_1,\tau_1} \ldots \mathbf{O}_{x_m,\tau_m} \rangle_J^c = \frac{1}{Z[J]} \int \mathcal{D}\phi \; \mathbf{O}_{x_1,\tau_1} \ldots \mathbf{O}_{x_m,\tau_m} \; e^{-S_0[\phi] - \int d^d x d\tau \, J_x \mathcal{O}_{x,\tau} - \int d^d x d\tau \; t_{x,\tau} \mathbf{O}_{x,\tau}} \Big|_{t=0}$$

$$= (-1)^m \frac{\partial}{\partial t_{x_1,\tau_1}} \frac{\partial}{\partial t_{x_2,\tau_2}} \cdots \frac{\partial}{\partial t_{x_m,\tau_m}} \ln \mathbf{Z}[J,t] \Big|_{t=0}. \tag{2.3}$$

It is a functional of random coupling $J_x$ as denoted by the subscript "$J$". The superscript "$c$" is a label for the connected correlation function. The generating functional of the connected correlation functions is thus defined as $\mathbf{G}[J,t] = \ln \mathbf{Z}[J,t]$. It is a functional of both external source $t_{x,\tau}$ and random coupling $J_x$. Consequently, the key to the disorder-averaged observables is to compute the disorder-averaged generating functional

$$\overline{\mathbf{G}}[W^2, t] = \int \mathcal{D}J \; P[J] \; \mathbf{G}[J,t] = \int \mathcal{D}J \; P[J] \; \ln \mathbf{Z}[J,t], \tag{2.4}$$

which leads to the disorder averaging of Eq. (2.3) being

$$\overline{\langle \mathbf{O}_{x_1,\tau_1} \mathbf{O}_{x_2,\tau_2} \ldots \mathbf{O}_{x_m,\tau_m} \rangle_J^c} = (-1)^m \frac{\partial}{\partial t_{x_1,\tau_1}} \frac{\partial}{\partial t_{x_2,\tau_2}} \cdots \frac{\partial}{\partial t_{x_m,\tau_m}} \overline{\mathbf{G}}[W^2, t] \Big|_{t=0}. \tag{2.5}$$

Conventionally, the replica trick simplifies the integration of a logarithm in Eq. (2.4), as reviewed below in Sec. II A. Alternatively, in this paper, we evaluate the disorder averaging in Eq. (2.4) by expanding the logarithm as a power series of the random coupling $J_x$ and then do the disorder averaging term by term. For a theory of generalized free fields, such as large N theories, if the disorder couples to the fundamental field, the series expansion gives a finite number of terms. Therefore, the disorder averaging in this case can be done exactly. In contrast, if the disorder couples to a composite operator of a free theory, or it couples to any operator in an interacting theory, the expansion leads to an infinite series. A truncation at finite order would be a good

approximation if the distribution of random coupling $J_x$ has small second and higher moments. For Gaussian distribution, this requires the width of the distribution $W^2$ to be small. More details will be given in Sec. II B.

## A.  Replica method

The replica trick is based on the identity $\ln \mathbf{Z} = \lim_{n\to 0} \frac{\mathbf{Z}^n - 1}{n}$. It gives $\overline{\mathbf{G}}[W^2, t] = \lim_{n\to 0} \frac{\overline{\mathbf{Z}^n}[t] - 1}{n}$, where

$$\overline{\mathbf{Z}^n}[t] = \int \mathcal{D}J \ P[J] \ \prod_a [\mathcal{D}\phi^a] \ e^{-\sum_{a=1}^n S_0^a[\phi^a] - \int d^dx \ J_x \ \int d\tau \sum_{a=1}^n \mathcal{O}_{x,\tau}^a - \int d^dx d\tau \sum_{a=1}^n t_{x,\tau}^a \mathbf{O}_{x,\tau}^a} \quad (2.6)$$

is the disorder averaged partition function of $n$-copies of the original system. Superscript $a$ is a replica index. If $P[J]$ is the Gaussian distribution given in Eq. (2.2), we can easily integrate over the random coupling $J_x$ and get

$$\overline{\mathbf{Z}^n}[t] = \int \prod_a [\mathcal{D}\phi^a] \ e^{-\sum_{a=1}^n S_0^a[\phi^a] + W^2 \int d^dx \left( \int d\tau \sum_{a=1}^n \mathcal{O}_{x,\tau}^a \right)^2 - \int d^dx d\tau \sum_{a=1}^n t_{x,\tau}^a \mathbf{O}_{x,\tau}^a}. \quad (2.7)$$

The disorder effect is fully encoded in the second term proportional to $W^2$. This term mixes operators from different replicas and is non-local along the time direction. Evidently, the effect of disorder at the clean fixed point is determined by whether this term is relevant or not. As $[x] = -1$, $[\tau] = -1$, and $[\mathcal{O}] = \Delta$, the scaling of randomness $W^2$ given by power counting is $[W^2] = d + 2 - 2\Delta$. Then, $[W^2] > 0$ ($[W^2] < 0$) indicates the disorder is (ir)relevant. In particular, if the disorder couples to the singlet operator with the lowest scaling dimension, since the critical exponent $\nu$ is associated with its scaling dimension $\Delta$ as $\nu = \frac{1}{d+1-\Delta}$, the dimension of the coupling $W^2$ can be reduced to $[W^2] = \frac{2}{\nu} - d$. This reproduces the familiar Harris criterion[34] that the disorder is (ir)relevant if $\frac{2}{\nu} \geq d$ ($\frac{2}{\nu} < d$). Therefore, the replicated theory gives a standard protocol to tell when the disorder is relevant no matter which operator the disorder couples to. Naturally, we are able to proceed to study the RG flow of the replicated theory. Upon Eq. (2.7), either momentum shell RG or field theoretical RG can be implemented following the standard procedure.

However, the replica method is valid only if the replicated action is bounded from below. Otherwise, it will give rise to instability. For example, as we will study in Sec. VI, the disorder averaging of $n$-copies of free bosonic theories with random mass, i.e. $S_0 = \int d^dx d\tau \ (\partial \phi_{x,\tau})^2$ and $\mathcal{O}_{x,\tau} = \phi_{x,\phi}^2$, generates a negative quartic term $-W^2 \int d^dx d\tau d\tau' \ \sum_{ab} (\phi_{x,\tau}^a)^2 (\phi_{x,\tau'}^b)^2$ in the effective action. Therefore, the replica approach cannot be used to study this particular disorder problem.

## B.   Traditional method by random coupling expansion

To avoid the issue caused by the replica trick, in this paper, we take another approach from the first principle. It was applied to the system of electrons scattered by random impurities decades ago[31–33]. Before we review the technical details of the method, the criterion for the relevant disorder is discussed. Without the replica trick, one can still tell if the disorder is relevant by the similar power counting. Here, the random coupling $J_x$ in the partition function Eq. (2.1) acquires a dimension $[J] = d + 1 - \Delta$ given $[\mathcal{O}] = \Delta$. This can directly lead to $[W^2] = d + 2 - 2\Delta$ in the probability density function of a Gaussian distribution. When $[W^2] > 0$, the Gaussian distribution is broadened under coarse graining, indicating more randomness is introduced to the system at lower energy scale and thus the disorder is relevant. When $[W^2] < 0$, the width of the Gaussian distribution becomes narrower. In the IR limit, the distribution is asymptotically a $\delta$-function, i.e. $\lim_{W^2 \to 0} P[J] = \prod_x \delta(J_x)$. As a result, the disorder averaging effectively imposes the constraint $J_x = 0$ at any site in the space, which leads to a clean theory in the long distance limit. In short, the disorder is irrelevant.

Instead of disorder averaging the UV action as we do in the replica method, here, we average the truncated series expansion of the generating functional. Namely,

$$\int \mathcal{D}J P[J] \, \mathbf{G}[J, t] \; \approx \; \sum_{\ell=0}^{L} \overline{J_{x_1} \ldots J_{x_\ell}} \, \mathcal{G}^{(\ell)}[t], \tag{2.8}$$

where we keep the first $L$ terms and $\mathcal{G}^{(\ell)}$ is the $\ell$-th coefficient. Later, in the study of the O(N) model, the random mass is drawn from a Gaussian distribution with zero mean and small variance suppressed by $\frac{1}{N}$. This gives nonzero $\overline{J_{x_1} \ldots J_{x_\ell}} \sim N^{-\frac{\ell}{2}}$ when $\ell$ is even. Then, the truncation error in Eq. (2.8) is of order $N^{-\frac{L}{2}}$ which can be ignored in the large N limit.

Diagrammatically, the computation of Eq. (2.8) including the series expansion and disorder averaging is represented in Eq. (2.9) and Eq. (2.10), successively. Together Eq. (2.8) with Eq. (2.3), the series expansion gives the m-point connected correlation function as

$$\langle \mathbf{O}_{x_1, \tau_1} \ldots \mathbf{O}_{x_m, \tau_m} \rangle_J^c = (-1)^m \sum_{\ell=0}^{L} \int d^d y_1 \ldots d^d y_\ell \left( \frac{\partial}{\partial t_{x_1, \tau_1}} \frac{\partial}{\partial t_{x_2, \tau_2}} \cdots \frac{\partial}{\partial t_{x_m, \tau_m}} \mathcal{G}^{(\ell)}[t] \right)_{t=0} J_{y_1} \ldots J_{y_\ell}$$

$$= \sum_{\ell=0}^{L} \int d^d y_1 \ldots d^d y_\ell \quad \text{} \quad . \tag{2.9}$$

The expression is truncated at the $L$-th order. There are $m$ external points in each diagram, denoted by black dots. Each red dot corresponds to a disorder coupling $J$. The $\ell$-th term contains all possible connected diagrams linking the $m$ external points with $\ell$ disorder couplings $J_{y_1} \ldots J_{y_\ell}$.

Then, for the Gaussian distribution, the disorder averaging is implemented in Eq. (2.10). Here, pairs of red dots are connected with a dashed line, according to the following rules.

$$\langle J_x J_{x'} \rangle = W^2 \underset{\text{•-----•}}{\delta^d(x-x')}, \qquad \text{or} \qquad \langle J_k J_{-k} \rangle = W^2 \underset{\text{•-----•}}{\delta(\omega)}. \qquad (2.10)$$

Only momentum, not frequency, is transferred along the dashed line. Later, we would study the vector O(N) model based on Eq. (2.9) and Eq. (2.10).

Conventional RG transformation can be implemented in the disordered theory. In the scheme of Wilsonian RG, we assign a hard UV cutoff $\Lambda$ to the system. As we integrate out fast modes with momentum in the shell $\Lambda - \delta\Lambda < k < \Lambda$ followed by a rescaling of spacetime to restore the UV cutoff, the form of the effective theory stays the same. This coarse graining induces the flow of couplings. The disordered coupling of $\mathcal{O}_{x,\tau}$ thus flows from $J_x$ to $J_x + \delta J_x$ with $\delta J_x \propto \delta\Lambda$. This means the random mass now has distribution $P'[J + \delta J]$ instead of original $P[J]$. For an infinitesimal RG step, the evolution of the Gaussian distribution is encoded in the change of its mean and variance $W^2$. Meanwhile, for a generic interacting theory, other couplings inevitably become functionals of $J_x$. This is due to the mixture of UV operators in which only the singlet couples to the disorder. This is the general picture of how a disordered theory flows under coarse graining without using the replica trick.

Besides the momentum shell RG, the field theoretical RG can also be formulated for an arbitrary theory coupled with disorder, with $\Lambda$ pushed to infinite. In the general theory in Eq. (2.1), one can define the renormalized field as $Z_{\mathbf{O}} \mathbf{O}_{R;x,\tau_R} = \mathbf{O}_{x,\tau}$ in the real space, where $\mathbf{O}_{x,\tau}$ can be $\phi_{x,\tau}$, $\mathcal{O}_{x,\tau}$, etc.. $Z_{\mathbf{O}}$ is a renormalization constant. Notice the quenched disorder gives rise to anisotropy in spacetime. We hence need to introduce renormalized time as $Z_\tau \tau_R = \tau$. The renormalized randomness is $Z_W W_R = W$ which controls the propability density function of the Gaussian disorder. Then, in terms of renormalized variables, the generating functional can be written as

$$\mathbf{G}[e_0; J, t] = \ln \int \mathcal{D}\phi_R \; e^{-S_0[e_0(e), Z_\phi \phi_{R;x,\tau_R}] - Z_\tau Z_{\mathcal{O}} \int d^d x d\tau_R J_x \mathcal{O}_{R;x,\tau_R} - Z_\tau Z_{\mathbf{O}} \int d^d x d\tau_R t_{x,\tau} \mathbf{O}_{R;x,\tau_R}}$$
$$= \mathbf{G}_R[e; Z_\tau Z_{\mathcal{O}} J, \; Z_\tau Z_{\mathbf{O}} t], \qquad (2.11)$$

where $e_0$ and $e$ collects all the bare and renormalized couplings in $S_0$. Equivalently, $\mathbf{G}_R[e; J, t] = \mathbf{G}[e_0; Z_\tau^{-1} Z_{\mathcal{O}}^{-1} J, Z_\tau^{-1} Z_{\mathbf{O}}^{-1} t]$. Then, the disorder averaged generating functional becomes

$$\overline{\mathbf{G}}[e_0, W^2, t] = \int \mathcal{D}J \; P[W^2, J] \; \mathbf{G}[e_0; J, t]$$
$$= \int \mathcal{D}\tilde{J} \; \tilde{P}[W_R^2, \tilde{J}] \; \mathbf{G}_R[e; \tilde{J}, Z_\tau Z_{\mathbf{O}} t]$$
$$= \overline{\mathbf{G}}_R[e, W_R^2, Z_\tau Z_{\mathbf{O}} t], \qquad (2.12)$$

where $\tilde{J}_x = Z_\tau Z_\mathcal{O} J_x$ and the Gaussian distribution parameterized by its variance $W^2$ is redefined as

$$\begin{aligned}
P[W^2, J] &= \frac{1}{\mathcal{N}} \exp\left\{ -\frac{1}{2W^2} \int d^d x J_x^2 \right\} \\
&= \frac{(Z_\tau Z_\mathcal{O})^V}{\tilde{\mathcal{N}}} \exp\left\{ -\frac{1}{2W_R^2} \int d^d x \tilde{J}_x^2 \right\} = (Z_\tau Z_\mathcal{O})^V \tilde{P}[W_R^2, \tilde{J}].
\end{aligned} \tag{2.13}$$

The new normalization factor is $\tilde{\mathcal{N}} = (Z_\tau Z_\mathcal{O})^V \mathcal{N}$. In addition, it is easy to find the following relation among the renormalization constants

$$Z_W^2 Z_\tau^2 Z_\mathcal{O}^2 = 1. \tag{2.14}$$

Then, the renormalized m-point correlation functions are given by

$$\begin{aligned}
\overline{\langle \mathbf{O}_{x_1,\tau_1} \dots \mathbf{O}_{x_m,\tau_m} \rangle_J^c} &= (-1)^m \frac{\delta \overline{\mathbf{G}}[e_0, W^2, t]}{\delta t_{x_1,\tau_1} \dots \delta t_{x_m,\tau_m}}\bigg|_{t=0} = (-1)^m \frac{\delta \overline{\mathbf{G}}_R[e, W_R^2, Z_\tau Z_\mathbf{O} t]}{\delta t_{x_1,\tau_1} \dots \delta t_{x_m,\tau_m}}\bigg|_{t=0} \\
&= Z_\tau^m Z_\mathbf{O}^m \overline{\langle \mathbf{O}_{R;x_1,\tau_{R;1}} \dots \mathbf{O}_{R;x_m,\tau_{R;m}} \rangle_J^c}.
\end{aligned} \tag{2.15}$$

In the momentum space, Fourier transformation gives $Z_\tau Z_\mathbf{O} \mathbf{O}_{R;k,\omega_R} = \mathbf{O}_{k,\omega}$. Besides, we have the renormalized frequency defined as $\omega = Z_\tau^{-1} \omega_R$. Then, the bare and renormalized correlation functions are related by

$$\overline{\langle \mathbf{O}_{k_1,\omega_1} \dots \mathbf{O}_{k_m,\omega_m} \rangle_J^c} \delta^d(\sum_{\ell=1}^m k_\ell) \delta(\sum_{\ell=1}^m \omega_\ell) = Z_\tau Z_\mathbf{O}^m \overline{\langle \mathbf{O}_{R;k_1,\omega_1} \dots \mathbf{O}_{R;k_m,\omega_m} \rangle_J^c} \delta^d(\sum_{\ell=1}^m k_\ell) \delta(\sum_{\ell=1}^m \omega_{R;\ell}) \tag{2.16}$$

All the above discussions can also be applied to disconnected correlation functions as well. They can be viewed as a product of connected correlation functions before disorder averaging. Consider a disconnected correlation function made up from $m$ connected correlation functions, the $k$-th of which has $n_k$ operators. Then, disorder averaging gives

$$\overline{\langle \prod_{i=1}^{n_1} \mathbf{O}_{x_{1_i},\tau_{1_i}} \rangle_J^c \dots \langle \prod_{j=1}^{n_m} \mathbf{O}_{x_{m_j},\tau_{m_j}} \rangle_J^c} = (Z_\tau Z_\mathbf{O})^{\sum_{j=1}^m n_j} \overline{\langle \prod_{i=1}^{n_1} \mathbf{O}_{R;x_{1_i},\tau_{R;1_i}} \rangle_J^c \dots \langle \prod_{j=1}^{n_m} \mathbf{O}_{R;x_{m_j},\tau_{R;m_j}} \rangle_J^c} \tag{2.17}$$

Diagramically, it is represented by a product of diagrams in Eq. (2.9). Each of them contains a fixed number of external points and disorder couplings denoted as red dots. Then, the disorder averaging connects all the red dots pairwise according to Eq. (2.10). This transfers momentum from a connected diagram to another. Thus, in the momentum space, there is one constraint for momenta while there are $m$ constraints for frequencies in the example of Eq. (2.17). In terms of the renormalized frequencies, different from Eq. (2.16), the constraints of the frequencies are written as

$$\prod_{\ell=1}^m \left[ \delta(\sum_{i=1}^{n_\ell} \omega_{\ell_i}) \right] = Z_\tau^m \prod_{\ell=1}^m \left[ \delta(\sum_{i=1}^{n_\ell} \omega_{R;\ell_i}) \right], \tag{2.18}$$

where each connected diagram gives a $Z_\tau$.

As usual, to the leading order of the perturbation, the renormalization constant determines the anomalous dimension. Particularly, operator $\mathbf{O}$ gains an anomalous dimension defined as $\gamma_{\mathbf{O}} = \mu\partial\mu \ln Z_{\mathbf{O}}$. Since $[W^2] = d + 2 - 2\Delta$, the $\beta$ function of $W_R^2$ is given by

$$\beta_W = -\mu\partial_\mu W_R^2 = (d + 2 - 2\Delta)W_R^2 + \gamma_W W_R^2 \tag{2.19}$$

where $\gamma_W = \mu\partial\mu \ln Z_W^2$. Similarly, $\mu\partial_\mu \ln Z_\tau$ plays the role of an anomalous dimension of time. Thus, we can characterize the spacetime anisotropy by the dynamical exponent defined as $z = 1 - \mu\partial_\mu \ln Z_\tau$.

## III.  RELEVANT DISORDER EFFECT ON THE O(N) VECTOR MODEL

From now on, we study the O(N) vector model perturbed by the random mass disorder. In other words, we introduce a random coupling to the singlet operator with the lowest scaling dimension. For a $d + 1$-dimensional system, the UV action in Euclidean spacetime is given by

$$S = \int d^d x d\tau \left[ \frac{1}{2}(\partial\phi_{x,\tau})^2 + \frac{1}{2}(m^2 + \frac{1}{\sqrt{N}}J_x')\phi_{x,\tau}^2 + \frac{\lambda}{2N}\phi_{x,\tau}^4 \right], \tag{3.1}$$

where $\phi_{x,\tau}$ is a $N$ component vector of real bosonic fields. Singlet operator $\phi_{x,\tau}^2 \equiv \sum_\alpha \phi_{x,\tau}^\alpha \phi_{x,\tau}^\alpha$ couples to a random coupling $J_x'$. $\alpha$ is the O(N) index and we will omit it throughout the paper for simplicity. $J_x'$ is drawn from a Gaussian distribution $P[J'] = \frac{1}{\mathcal{N}_0} \exp\{-\frac{1}{2W_0^2} \int d^d x (J_x')^2\}$ with zero mean and variance $W_0^2$. The normalization factor is $\mathcal{N}_0 = (2\pi W_0^2)^{V/2}$. We suppress the disorder coupling by a factor of $\frac{1}{\sqrt{N}}$. In the action, the quartic term is (ir)relevant at $d < 3$ ($d > 3$). Therefore, we will separately discuss the effect of the disorder in $d < 3$ and $d > 3$ spatial dimensions.

### A.  $d < 3$

In this case, it is well-known that, the quartic interaction is relevant in the clean system which drives the RG flow from the UV Gaussian fixed point towards the stable Wilson-Fisher fixed point when $m^2 = 0$. In order to write the fixed point action in the IR, it is convenient to do the Hubbard-Stratonovich transformation by introducing an auxiliary field $\sigma_{x,\tau}$. In the presence of the disorder, we follow the same procedure, which gives

$$S = \int d^d x d\tau \left[ \frac{1}{2}(\partial\phi_{x,\tau})^2 + \frac{1}{2}m^2\phi_{x,\tau}^2 + \frac{1}{2}i\sigma_{x,\tau}\phi_{x,\tau}^2 + \frac{N}{4\lambda}[\sigma_{x,\tau} + i\frac{1}{\sqrt{N}}J_x']^2 \right]$$

$$= \int d^d x d\tau \left[ \frac{1}{2}(\partial\phi_{x,\tau})^2 + \frac{1}{2}m^2\phi_{x,\tau}^2 + \frac{1}{2}i\sigma_{x,\tau}\phi_{x,\tau}^2 + i\sqrt{N}J_x\sigma_{x,\tau} + \frac{N}{4\lambda}\sigma_{x,\tau}^2 - \lambda J_x^2 \right]. \tag{3.2}$$

In the second equality, we redefine $J_x = \frac{1}{2\lambda}J'_x$. Correspondingly, the Gaussian distribution for $J_x$ has width $W^2 = W_0^2/(4\lambda^2)$ and the normalization becomes $\mathcal{N} = \mathcal{N}_0/(2\lambda)^V$. The last term is a constant and can be dropped. As the mass is fine tuned to zero, without considering the irrelevant term $\sigma_{x,\tau}^2$, we can get the critical action of the Wilson-Fisher fixed point coupled to disorder as

$$S = \int d^dx d\tau \left[ \frac{1}{2}(\partial\phi_{x,\tau})^2 + \frac{1}{2}i\sigma_{x,\tau}\phi_{x,\tau}^2 + i\sqrt{N}J_x\sigma_{x,\tau} \right]. \tag{3.3}$$

Here, the random mass disorder is coupled with the singlet field $\sigma$. The bosonic fields $\phi$ and $\sigma$ have engineer dimensions $\Delta_\phi = \frac{d-1}{2}$ and $\Delta_\sigma = 2$. Thus, the relevance of disorder requires $d + 2 - 2\Delta_\sigma > 0$, i.e. $d > 2$.

## B.   $d > 3$

In this case, the quadratic action

$$S = \int d^dx d\tau \left[ \frac{1}{2}(\partial\phi_{x,\tau})^2 + \frac{1}{2}(m^2 + \frac{1}{\sqrt{N}}J'_x)\phi_{x,\tau}^2 \right] \tag{3.4}$$

is enough to describe the system. By fine tuning $m^2 = 0$ in the absence of disorder, we can approach the Gaussian fixed point where the free field has scaling dimension $\Delta_\phi = \frac{d-1}{2}$. This means the operator coupled to the disorder has the scaling dimension $\Delta_{\phi^2} = d - 1$. When $d + 2 - 2\Delta_{\phi^2} > 0$, i.e. $d < 4$, the disorder is relevant.

## IV.   THE O(N) VECTOR MODEL AT $N = \infty$ IN $2 < d < 3$ SPATIAL DIMENSIONS

In this section, we study the $N = \infty$ disordered O(N) vector model in Eq. (3.3) which is of purely theoretical interest. In this case, the Wilson-Fisher fixed point is described by two generalized free fields [2] : the free bosonic field $\phi$ and the singlet field $\sigma$ with their scaling dimensions equal to engineer dimensions. The coupling between them is of order $1/N$. Thus, as $N \to \infty$, the two sectors of operators $\{\phi^n\}$ and $\{\sigma^n\}$, made up of $\phi$, $\sigma$ and their composite operators respectively, are decoupled. The disorder couples to $\sigma$ and hence leaves the $\phi$ sector intact. We can then integrate over the $\phi$ field in Eq. (3.3). After redefining the field as $\sigma_{x,\tau} \to \frac{1}{\sqrt{N}}\sigma_{x,\tau}$, the resulting effective action is

$$S_{N=\infty}^{\text{WF}} = \frac{1}{2}\int d^dk d\omega \, G_\sigma^{-1}(k,\omega)\sigma_{k,\omega}\sigma_{-k,-\omega} + i\int d^dk d\omega \, J_{-k}\delta(\omega)\sigma_{k,\omega}, \tag{4.1}$$

where the propagator of $\sigma$ in the momentum space is $G_\sigma(k,\omega) = \frac{2}{c_2}(k^2 + \omega^2)^{\frac{3-d}{2}}$ with $c_2 = -\frac{2^{1-2d}\pi^{\frac{2-d}{2}}}{\Gamma(\frac{d}{2})\sin\frac{\pi(d+1)}{2}} > 0$, as computed in Appendix A.

---

[2] Generalized free field is one of the CFT operators that has scaling dimension $\Delta$ different from that of free fields, i.e. $\Delta \neq \frac{d-1}{2}$. It has non-zero two point function with the form $\frac{1}{r^{2\Delta}}$ and vanishing higher point functions. The theory for single generalized free field $\Phi$ has Gaussian type of action $S = \int d^{d+1}r \, \Phi(-\nabla^2)^{\Delta-\frac{d+1}{2}}\Phi$.

## A.  Disorder averaged connected correlation functions

Making use of the Feynman rules in Eq.(2.9) and Eq. (2.10), we can obtain the diagrammatic expression of the disordered averaged connected correlation functions, by listing all the connected diagrams before directly linking the random couplings with dashed lines. In the following, a few expressions are given.

$$\overline{\langle \sigma_{x_1,\tau_1} \sigma_{x_2,\tau_2} \rangle_J^c} = \text{\textasciitilde\textbullet} = \int \frac{d^d k d\omega}{(2\pi)^{d+1}} \, G_\sigma(k,\omega) \, e^{ik\cdot(x_1-x_2)+i\omega(\tau_1-\tau_2)}, \tag{4.2}$$

$$\overline{\langle \sigma_{x_1,\tau_1}^2 \sigma_{x_2,\tau_2}^2 \rangle_J^c} = 2 \, \text{\textbackslash diagram} + 4 \, \text{\textbackslash diagram}, \tag{4.3}$$

$$\overline{\langle \sigma_{x_1,\tau_1}^2 \sigma_{x_2,\tau_2}^2 \sigma_{x_3,\tau_3}^2 \rangle_J^c} = 8 \, \triangle + 8 \Big( \triangle + \triangle + \triangle \Big). \tag{4.4}$$

As a generalized free field, the connected higher-point correlation functions of $\sigma$ are all zero. For composite operators of $\sigma$, higher-point functions are nonzero and can be obtained in the same way. With a finite number of external points, there are always a finite number of connected diagrams which contribute to the correlation functions.

Notice disorder doesn't affect the connected 2-point function of $\sigma_{x,\tau}$. However, if we compute its disconnected 2-point function, the correction by disorder appears, as denoted below by $G_d$. Namely,

$$\overline{\langle \sigma_{x_1,\tau_1} \sigma_{x_2,\tau_2} \rangle_J} = \overline{\langle \sigma_{x_1,\tau_1} \sigma_{x_2,\tau_2} \rangle_J^c} + G_d, \tag{4.5}$$

where

$$G_d(k,\omega) = \text{\textasciitilde\textbullet----\textbullet\textasciitilde} = -W^2 [G_\sigma(k,0)]^2 \delta(\omega) = -\frac{4W^2}{c_2^2} k^{2(3-d)} \delta(\omega). \tag{4.6}$$

It contains two connected diagrams if we cut the dashed line. In the real space, the disorder effect is given by

$$G_d(x,\tau) = \int \frac{d^d k}{(2\pi)^d} \frac{d\omega}{2\pi} G_d(k,\omega) e^{ik\cdot x + i\omega\tau} = -\frac{4W^2}{c_2^2} \int \frac{d^d k}{(2\pi)^d} \, k^{2(3-d)} \, e^{ik\cdot x} = -\frac{W^2 c_1}{|x|^{6-d}}, \tag{4.7}$$

where $c_1 = \frac{2^{2d+6}\pi^{\frac{d}{2}-2}\cos^2[\frac{\pi}{2}d]\Gamma[3-\frac{d}{2}]\Gamma^2[\frac{d}{2}]}{\Gamma[d-3]}$. It is independent of temporal separation. Accordingly, the connected Green's functions of $\sigma_{x,\tau}^2$ are

$$\overline{\langle \sigma_{x_1,\tau_1}^2 \sigma_{x_2,\tau_2}^2 \rangle_J^c} = 2c_3^2 (x_{12}^2 + \tau_{12}^2)^{-4} - 4c_1 c_3 W^2 (x_{12}^2 + \tau_{12}^2)^{-2} (x_{12}^2)^{\frac{d-6}{2}}, \tag{4.8}$$

$$\overline{\langle \sigma_{x_1,\tau_1}^2 \sigma_{x_2,\tau_2}^2 \sigma_{x_3,\tau_3}^2 \rangle_J^c} = 8c_3^3 (x_{12}^2 + \tau_{12}^2)^{-2} (x_{13}^2 + \tau_{13}^2)^{-2} (x_{23}^2 + \tau_{23}^2)^{-2}$$
$$- 8c_1 c_3^2 W^2 \left[ (x_{12}^2 + \tau_{12}^2)^{-2} (x_{13}^2 + \tau_{13}^2)^{-2} (x_{23}^2)^{\frac{d-6}{2}} + \text{permutations} \right], \tag{4.9}$$

where $x_{ij} = x_i - x_j$ and $\tau_{ij} = \tau_i - \tau_j$. The coefficient $c_3 = -\frac{2^{d+3}\cos[\frac{\pi}{2}d]\Gamma[\frac{d}{2}]}{\pi^{\frac{3}{2}}\Gamma[\frac{d-3}{2}]}$ is negative at $2 < d < 3$. These results are reproduced by exact computation in Appendix B or by using the conventional replica method in Appendix C. These two correlation functions have power law behavior. Notice at integer dimensions, the disorder doesn't affect the system due to $c_1 = 0$ even though it is relevant at $d > 2$. At fractional dimensions, the correction proportional to $W^2$ is finite.

It is believed that all the correlation functions obey the power law. We now study if there is an appropriate scale transformation such that the theory in the IR limit is scale invariant. As shown in Eq. (4.2), the 2-point function of $\sigma$ is the same as in the clean theory. Thus, under scale transformation $x_{ij} \to x_{ij}e^\ell$ and $\tau_{ij} \to \tau_{ij}e^\ell$, $\overline{\langle \sigma_{x_1,\tau_1}\sigma_{x_2,\tau_2}\rangle^c_J}$ is invariant if $\sigma$ is a scaling operator and has scaling dimension $\Delta_\sigma = 2$. Here, $\ell$ is the change in logarithmic length scale. It remains positive as we approach to the IR limit. Meanwhile, this transformation of spacetime makes it evident that the contribution proportional to $W^2$ in Eq. (4.8) and Eq. (4.9) dominants in the long distance limit when $d > 2$ as long as $c_1 \neq 0$. As a result, these correlation functions are approximately

$$e^{8\ell}e^{-(d-2)\ell}\overline{\langle \sigma^2_{x_1,\tau_1}\sigma^2_{x_2,\tau_2}\rangle^c_J} \approx -4c_1 c_3 W^2 (x_{12}^2 + \tau_{12}^2)^{-2}(x_{12}^2)^{\frac{d-6}{2}}, \tag{4.10}$$

$$e^{12\ell}e^{-(d-2)\ell}\overline{\langle \sigma^2_{x_1,\tau_1}\sigma^2_{x_2,\tau_2}\sigma^2_{x_3,\tau_3}\rangle^c_J} \approx -8c_1 c_3^2 W^2 \Big[(x_{12}^2 + \tau_{12}^2)^{-2}(x_{13}^2 + \tau_{13}^2)^{-2}(x_{23}^2)^{\frac{d-6}{2}}$$

$$+ \text{permutations}\Big]. \tag{4.11}$$

On the left hand side, the scale transformation $\sigma^2 \to e^{-\Delta^{d(2\text{pt})}_{\sigma^2}\ell}\sigma^2$ with $\Delta^{d(2\text{pt})}_{\sigma^2} = 4 - \frac{d-2}{2}$ leaves the 2-point function invariant. When $d > 2$, $\Delta^{d(2\text{pt})}_{\sigma^2} \neq \Delta_{\sigma^2} = 2\Delta_\sigma$. In other words, the composite operator $\sigma^2$ acquires an anomalous dimension $\gamma_{\sigma^2} = 1 - \frac{d}{2}$ which only vanishes at $d = 2$. For the 3-point function, the scale transformation of $\sigma^2$ becomes $\sigma^2 \to e^{-\Delta^{d(3\text{pt})}_{\sigma^2}\ell}\sigma^2$ where $\Delta^{d(3\text{pt})}_{\sigma^2} = 4 - \frac{d-2}{3} \neq \Delta^{d(2\text{pt})}_{\sigma^2}$. Moreover, we can consider an $m$-point function of $\sigma^2$, where $\sigma^2$ acquires a scaling dimension $\Delta^{d(m\text{pt})}_{\sigma^2} = 4 - \frac{d-2}{m}$. Consequently, the scaling operator $\sigma^2$ acquires an infinite number of distinct scaling dimensions $\Delta^{d(2\text{pt})}_{\sigma^2} \neq \Delta^{d(m\text{pt})}_{\sigma^2}$ with $m > 2$, which is impossible in a well-defined scale invariant theory. For other composite operators, this inconsistency also exists.

Naturally, we would like to fix the scaling dimension of $\sigma^2$ to be $\Delta^{d(2\text{pt})}_{\sigma^2}$. Then, in the IR, all its higher-point functions vanish. In this way, we can get a scale invariant theory where each composite operator in the clean system acquires a distinct anomalous dimension, as listed in Tab. I. This is like a theory of infinitely many generalized free fields but no composite of them.

Alternatively, it is more interesting to introduce an intrinsic scale set by the dimensionful coupling $W^2$. In other words, under scale transformation, $W^2$ is also changed as $W^2 \to W^2 e^{-\Delta_{W^2}\ell}$ where $\Delta_{W^2} = d + 2 - 2\Delta_\sigma = d - 2$. Then, the composite operator $\sigma^n$ can be assigned a scaling dimension $\Delta_{\sigma^n} = 2n$ the same as in the clean theory. This disordered theory with an intrinsic scale is analogous to the theory of fermi surface, where the fermi momentum plays the role of intrinsic scale. The coarse graining effectively increases the fermi momentum in the long distance

| operators | $\Delta_{\text{clean}}$ | $\Delta_{\text{dirty}}$ |
|:---:|:---:|:---:|
| $\sigma$ | $\Delta_\sigma$ | $\Delta_\sigma$ |
| $\sigma^2$ | $2\Delta_\sigma$ | $2\Delta_\sigma - \frac{d-2}{2}$ |
| $\sigma^3$ | $3\Delta_\sigma$ | $3\Delta_\sigma - 2\frac{d-2}{2}$ |
| $\vdots$ | $\vdots$ | $\vdots$ |
| $\sigma^n$ | $n\Delta_\sigma$ | $n\Delta_\sigma - (n-1)\frac{d-2}{2}$ |
| $\vdots$ | $\vdots$ | $\vdots$ |

TABLE I. The clean system consists operator $\sigma$ and its composite operators. The latter has scaling dimension equal to a multiple of the scaling dimension of $\sigma$, i.e. $\Delta_\sigma$. In the disordered systems, except the operator $\sigma$, all other operators gain anomalous dimensions listed in the third column.

limit. Similarly, in the system we study here, the disorder strength $W^2$ increases when $d > 2$ as we approach the IR limit. As will be studied in Sec. IV B, normal to the direction of the intrinsic scale, the subdimensional system is scale, even conformal, invariant. This is the phenomenon of dimension reduction which has been found and studied extensively in the effective theory of the random field Ising model[35–41].

Although the space and time transform in the same way to preserve the scale invariance as we discussed above, i.e. the dynamical exponent satisfies $z = 1$, the behavior of correlation functions has anisotropy in space and time. For example, the equal time and almost equal space 2-point correlation functions are

$$\overline{\langle \sigma^2_{x_1,0} \sigma^2_{x_2,0} \rangle^c_J} \approx -4c_1 c_3 W^2 |x_{12}|^{d-10}, \tag{4.12}$$

$$\overline{\langle \sigma^2_{0,\tau_1} \sigma^2_{a,\tau_2} \rangle^c_J} \approx -4c_1 c_3 W^2 |\tau_{12}|^{-4} a^{d-6} \tag{4.13}$$

in the long distance limit. In Eq. (4.13), we set the spatial splitting $a$ very small so that $0 < a = |x_{12}| \ll |\tau_{12}|$.

## B. Dimension reduction

As we pointed out, the critical $O(N)$ model in $2 < d < 3$ dimensions at $N = \infty$ can be understood as a theory with an intrinsic scale. In this section, we would like to make it clear in its effective theory. In order to do this, we introduce fermions into our system and rewrite our theory. Starting with the free disordered theory in Eq. (4.1), any disorder averaged observable can

be computed as

$$\overline{\langle (\mathbf{O}_{x_1,\tau_1}\ldots\mathbf{O}_{x_m,\tau_m})[\sigma]\rangle_J^c} = \int \mathcal{D}J\; P[J]\; (Z[J])^{-1} \int \mathcal{D}\sigma\; (\mathbf{O}_{x_1,\tau_1}\ldots\mathbf{O}_{x_m,\tau_m})[\sigma]\; e^{-S_{N=\infty}^{\mathrm{WF}}[\sigma;J]}$$

$$= \int \mathcal{D}\sigma\; (\mathbf{O}_{x_1,\tau_1}\ldots\mathbf{O}_{x_m,\tau_m})[\sigma]\; e^{-\overline{S_{\mathrm{eff}}[\{\sigma_{x,\tau}\}]}}, \qquad (4.14)$$

where each operator $\mathbf{O}_{x,\tau}$ is a functional of the fundamental operator $\sigma_{x,\tau}$. The disorder averaged effective action is thus

$$e^{-\overline{S_{\mathrm{eff}}[\{\sigma_{x,\tau}\}]}} = \int \mathcal{D}J\; P[J]\; (Z[J])^{-1} \int \mathcal{D}\sigma\; e^{-S_{N=\infty}^{\mathrm{WF}}[\sigma;J]}$$

$$= \sqrt{\left(\frac{W^2}{2\pi}\right)^V} \int \mathcal{D}\psi\mathcal{D}\bar\psi\mathcal{D}\xi\; e^{-S_{\mathrm{eff}}'[\sigma,\xi,\psi,\bar\psi]}. \qquad (4.15)$$

where $\psi$ and $\bar\psi$ are anti-commuting fields while $\xi$ is a commuting field as $\sigma$. Given

$$(Z[J])^{-1} = \sqrt{(2\pi)^{-V} \prod_{k,\omega} G_\sigma^{-1}(k,\omega)}\; \exp\left(\frac{1}{2}\int d^d k\; G_\sigma(k,0) J_k J_{-k}\right), \qquad (4.16)$$

we can get

$$S_{\mathrm{eff}}'[\sigma,\xi,\psi,\bar\psi] = \int d^d k d\omega\; \Big[\frac{1}{2}G_\sigma^{-1}(k,\omega)\xi_{k,\omega}\xi_{-k,-\omega} + \bar\psi_{k,\omega}G_\sigma^{-1}(k,\omega)\psi_{-k,-\omega}$$

$$+ \frac{1}{2}G_\sigma^{-1}(k,\omega)\sigma_{k,\omega}\sigma_{-k,-\omega}\Big] - \frac{W^2}{2}\int d^d k\; (\xi_{k,0} - i\sigma_{k,0})(\xi_{-k,0} - i\sigma_{-k,0}), \quad (4.17)$$

after integrating out the Gaussian disorder. This effective action can be simplified by defining new commuting complex fields $\eta_{k,\omega} = \xi_{k,\omega} - i\sigma_{k,\omega}$ and $\varphi_{k,\omega} = \frac{1}{2}(\xi_{k,\omega} + i\sigma_{k,\omega})$, which gives

$$S_{\mathrm{eff}}[\eta,\varphi,\psi,\bar\psi] = \int d^d k d\omega\; G_\sigma^{-1}(k,\omega)\Big[\eta_{k,\omega}\varphi_{-k,-\omega} + \bar\psi_{k,\omega}\psi_{-k,-\omega}\Big] - \frac{W^2}{2}\int d^d k\; \eta_{k,0}\eta_{-k,0}. \quad (4.18)$$

It can be used to compute any disorder averaged correlation function of operators rewritten in terms of the $\eta$ and $\varphi$. The disorder only couples to the $\eta$ field at zero frequency $\omega = 0$. So we can integrate out the fields at nonzero frequency. Recall $G_\sigma(k,\omega=0) = \frac{2}{c_2}(k^2)^{\frac{3-d}{2}}$. Without loss of generality, we can set $\frac{2W^2}{c_2} = 1$ and the effective action becomes

$$\frac{2}{c_2} S_{\mathrm{eff}}^{\omega=0}[\eta,\varphi,\psi,\bar\psi] = \int d^d k\; \Bigg[ (k^2)^{\frac{d-3}{2}}\eta_{k,0}\varphi_{-k,0} + (k^2)^{\frac{d-3}{2}}\bar\psi_{k,0}\psi_{-k,0} - \frac{1}{2}\eta_{k,0}\eta_{-k,0}\Bigg]. \qquad (4.19)$$

This theory can also be obtained using the replica trick as we show in Appendix D and the physical meaning of these fields is rather obvious in terms of replica fields. Suppose the theory is isotropic, which means the Lagrangian is independent of the directions of momenta. Then, with $\int d^d k = \Omega_d \int k^{d-1} dk$ where $\Omega_d = \frac{2\pi^{d/2}}{\Gamma[d/2]}$, we can define a new momentum $p_\mu$ in $\mathfrak{D} = \frac{2d}{d+1-2\Delta_\sigma}$ dimensions satisfying $p^2 \equiv p_\nu p^\nu = (k_\mu k^\mu)^{\frac{d-3}{2}}$, such that the original theory can be viewed as a free

theory of fields $\eta'_p = \eta_{k,0}$, $\bar{\psi}'_p = \bar{\psi}_{k,0}$, $\psi'_p = \psi_{k,0}$ and $\varphi'_p = \varphi_{k,0}$ in $\mathfrak{D} = \frac{2d}{d-3}$ dimensions. The effective action is hence in the following familiar form:

$$
\begin{aligned}
\frac{\Omega_{\mathfrak{D}}}{\Omega_d} \frac{2}{c_2} \mathcal{S}_{\text{eff}}^{\omega=0}[\eta', \varphi', \psi', \bar{\psi}'] &= \int d^{\mathfrak{D}}p \left[ p^2 \eta'_p \varphi'_{-p} + p^2 \bar{\psi}'_p \psi'_{-p} - \frac{1}{2} \eta'_p \eta'_{-p} \right] \\
&= \int d^{\mathfrak{D}}x \left[ \eta'_x(-\nabla^2)\varphi'_x + \bar{\psi}'_x(-\nabla^2)\psi'_x - \frac{1}{2}(\eta'_x)^2 \right].
\end{aligned}
\tag{4.20}
$$

Previous study[37] tells us that this action has a supersymmetry whose transformation is

$$
\delta\varphi'_x = -\bar{a}\epsilon_\mu x_\mu \psi'_x, \quad \delta\eta'_x = 2\bar{a}\epsilon_\mu \partial_\mu \psi'_x, \quad \delta\psi'_x = 0, \quad \delta\bar{\psi}'_x = \bar{a}\epsilon_\mu(x_\mu \eta'_x + 2\partial_\mu \varphi'_x),
\tag{4.21}
$$

where $\bar{a}$ is an infinitesimal anticommuting number and $\epsilon_\mu$ is an arbitrary vector. Then, we can define a superfield $\Phi_x = \varphi'_x + \theta\psi'_x + \bar{\theta}\bar{\psi}'_x + \theta\bar{\theta}\eta'_x$ such that the effective action in Eq. (4.20) can be written in a rather simple form as

$$
\mathcal{S}_{\text{eff}}^{\omega=0}[\Phi] = c_2 \frac{\Omega_d}{\Omega_{\mathfrak{D}}} \int d^{\mathfrak{D}}x d\theta d\bar{\theta} \left( \Phi_x[-\nabla^2 - 2\partial_{\bar{\theta}}\partial_\theta]\Phi_x \right),
\tag{4.22}
$$

where the Berezin integral over the Grassmann variable $\theta$ or $\bar{\theta}$ is defined to be $\int d\theta\theta = \int d\bar{\theta}\bar{\theta} = 1$. The Grassmannian coordinates have integral negative dimensions. Therefore, this theory describes a free CFT in $\mathfrak{D} - 2 = \frac{4\Delta_\sigma - 2}{d-3}$ dimensions. This means the original $d + 1$ spacetime dimensional system has the scale invariance in $2\Delta_\sigma - 1 = 3$ subdimensional system. Interestingly, when $d = 2$, the total spacetime dimension saturates this requirement. So the disordered system is scale invariant in the IR. While when $d > 2$, apart from the scale invariant three dimensional system, there is extra fractional dimension $d-2$ left as the dimension of the intrinsic scale. Finally, I would like to briefly comment on what happens at $d = 3$. Notice the operator $\sigma^2$ is exactly marginal. So, it should be included in the action in Eq. (4.1). Consequently, $G_\sigma^{-1}(k, \omega)$ reduces to a constant. Then, the effective action defined in Eq. (4.15) by integrating over random coupling $J$ is only a functional of the bosonic field $\sigma$. At $\omega = 0$, it describes a scale invariant theory in $d = 3$ dimensions although the whole system lives in $d + 1 = 4$ dimensions.

## V.    $1/N$ **CORRECTION AT** $d = 2 + \epsilon$

In this section, we study the critical O(N) vector model at $d = 2 + \epsilon$ and finite $N$ using double expansion of $\epsilon$ and $1/N$. The bare action is

$$
\begin{aligned}
S_B = {}& \frac{1}{2} \int d^dx d\tau \, (\partial\phi_{x,\tau})^2 + \frac{1}{2} \int d^dx d\tau \int d^dx' d\tau' \, \sigma_{x,\tau} G_\sigma^{-1}(|x - x'|, |\tau - \tau'|)\sigma_{x',\tau'} \\
& + \frac{i}{2\sqrt{N}} \int d^dx d\tau \, \sigma_{x,\tau}\phi_{x,\tau}^2 + i \int d^dx d\tau \, J_x \sigma_{x,\tau}.
\end{aligned}
\tag{5.1}
$$

And the Gaussian disorder has a bare distribution $P[J] = \frac{1}{\mathcal{N}} \exp\left\{-\frac{1}{2W^2} \int d^d x J_x^2\right\}$. The propagator of $\sigma$ is $G_\sigma(k, \omega) = \frac{2}{c_2}[k^2 + \omega^2]^{\frac{1}{2}}$ in $2+1$ dimensions. Accordingly, the Feynman rules can be defined as

$$G_\phi = \frac{1}{k^2 + \omega^2} \quad, \qquad \frac{i}{2\sqrt{N}} \quad, \qquad G_\sigma = \frac{2}{c_2}[k^2 + \omega^2]^{\frac{1}{2}} \quad, \qquad G_d = -\frac{4W^2}{c_2^2} k^2 \delta(\omega) \quad. \tag{5.2}$$

At the order of $1/N$, the bare connected correlation functions $G_\phi^B$ and $G_\sigma^B$ acquire corrections as shown in the following.

$$G_\phi^B = \quad + \quad L_1 \quad + \quad L_2 \quad + \quad \mathcal{O}\left(\frac{1}{N^2}\right),$$

$$G_\sigma^B = \quad + \quad L_3 \quad + \quad L_4 \quad + \quad L_5 \quad + \quad L_6$$

$$+ \quad L_7 \quad + \quad L_8 \quad + \quad \mathcal{O}\left(\frac{1}{N^2}\right). \tag{5.3}$$

The loop diagrams are evaluated in Appendix E. The results are listed in Tab. II, which are consistent with [6].

| $L_1(k,\omega)$ | $\frac{8}{3\pi^2 N}\frac{1}{\epsilon}(k^2 + \omega^2)\mu^\epsilon$ | $L_2(k,\omega)$ | $\frac{W^2}{N}\frac{128}{\pi}\frac{1}{\epsilon}(\omega^2 - k^2)\mu^\epsilon$ |
|---|---|---|---|
| $L_3(k,\omega)$ | $-\frac{1}{3\pi^2 N}\frac{\mu^\epsilon}{\epsilon}\frac{1}{(k^2+\omega^2)^{1/2}}$ | $L_4(k,\omega)$ | $\frac{8W^2}{\pi N}\frac{\mu^\epsilon}{\epsilon}\frac{k^2}{(k^2+\omega^2)^{\frac{3}{2}}}$ |
| $L_5(k,\omega)$ | $-\frac{1}{\pi^2 N}\frac{\mu^\epsilon}{\epsilon}[k^2 + \omega^2]^{-\frac{1}{2}}$ | $L_6(k,\omega)$ | $\frac{16W^2}{\pi N}\frac{\mu^\epsilon}{\epsilon}[k^2 + \omega^2]^{-\frac{1}{2}}$ |

TABLE II. Evaluation of the UV divergence of loop diagrams $L_1 \sim L_6$ at the order of $1/N$. $\mu$ is the renormalization scale. $L_7$ and $L_8$ are not divergent.

In total, the bare correlation functions are

$$G_\phi^B(k, \omega) = \frac{1}{(k^2 + \omega^2)} + \frac{L_1 + L_2}{(k^2 + \omega^2)^2},$$

$$G_\sigma^B(k, \omega) = 16[k^2 + \omega^2]^{\frac{1}{2}} + 256[k^2 + \omega^2](L_3 + L_4 + L_5 + L_6). \tag{5.4}$$

In terms of the renormalized variables defined as

$$\phi_R = Z_\phi^{-1}\phi, \quad \sigma_R = Z_\sigma^{-1}\sigma, \quad \tau = Z_\tau \tau_R, \quad \omega = \omega_R/Z_\tau, \quad Z_W W_R = W, \tag{5.5}$$

we can obtain the finite renormalized connected correlation functions $G_\phi^R(k, \omega_R) = Z_\phi^{-2} Z_\tau^{-1} G_\phi^B(k, \omega)$ and $G_\sigma^R(k, \omega_R) = Z_\sigma^{-2} Z_\tau^{-1} G_\sigma^B(k, \omega)$. The divergence in the bare correlation function can be canceled by counterterms, which fixes the renormalization constants to be

$$Z_\tau = 1 - \frac{W_R^2}{N} \frac{128}{\pi} \frac{\mu^\epsilon}{\epsilon}, \tag{5.6}$$

$$Z_\phi = 1 + \frac{1}{2} \left( \frac{8}{3\pi^2 N} \right) \frac{\mu^\epsilon}{\epsilon}, \tag{5.7}$$

$$Z_\sigma = 1 + \frac{1}{2} \left( -\frac{64}{3\pi^2 N} + \frac{512 W_R^2}{\pi N} \right) \frac{\mu^\epsilon}{\epsilon}. \tag{5.8}$$

Then, the anomalous dimensions of $\phi$ and $\sigma$ operators as well as the dynamical exponent are determined as

$$\gamma_\phi = \mu \partial_\mu \ln Z_\phi = \frac{1}{2} \left( \frac{8}{3\pi^2 N} \right), \tag{5.9}$$

$$\gamma_\sigma = \mu \partial_\mu \ln Z_\sigma = \frac{1}{2} \left( -\frac{64}{3\pi^2 N} + \frac{512 W_R^2}{\pi N} \right), \tag{5.10}$$

$$z = 1 - \mu \partial_\mu \ln Z_\tau = 1 + \frac{W_R^2}{N} \frac{128}{\pi}. \tag{5.11}$$

We can also study the renormalization of the composite operators. At the order of $1/N$, the operator $\sigma^2$ acquires an anomalous dimension twice as that of $\sigma$. The equality in Eq. (2.14) further gives

$$Z_W = 1 + \frac{1}{2} \left( \frac{64}{3\pi^2} - \frac{256 W_R^2}{\pi} \right) \frac{1}{N} \frac{\mu^\epsilon}{\epsilon}, \tag{5.12}$$

which gives the $\beta$ function of the randomness $W_R^2$ as

$$\beta_{W^2} = -\frac{dW_R^2}{d \ln \mu} = \epsilon W_R^2 + W_R^2 \frac{d \ln Z_W^2}{d \ln \mu} = \epsilon W_R^2 + \frac{64 W_R^2}{3\pi^2 N} - \frac{256 W_R^4}{\pi N}. \tag{5.13}$$

Besides the unstable clean fixed point at $W_R^2 = 0$, there is another fixed point at $\frac{W_R^2}{N} = \frac{\pi}{256} (\epsilon + \frac{64}{3\pi^2 N})$ [3]. At this disordered fixed point, there are universal quantities

$$\gamma_\sigma^* = \epsilon + \frac{32}{3\pi^2 N}, \tag{5.14}$$

$$z^* = 1 + \frac{\epsilon}{2} + \frac{32}{3\pi^2 N}. \tag{5.15}$$

More discussion will be given in Sec. VII on the fixed point structure in the space of randomness $W_R^2$.

---

[3] $W_R^2/N$ is the small parameter in the perturbative study.

## VI.   $1/N$ CORRECTION AT $d = 4 - \epsilon$

### A.   Field theoretical RG

At $3 < d < 4$, as analyzed before in Sec. III B, we consider the perturbation around the Gaussian fixed point by disorder, where the disorder couples to the composite operator $\phi^2$ with scaling dimension $\Delta_{\phi^2} = d - 1$. The action in Eq. (3.4) at $m = 0$

$$S = \frac{1}{2} \int d^d x \, d\tau [(\partial \phi_{x,\tau})^2 + \frac{1}{\sqrt{N}} J_x \phi_{x,\tau}^2] \tag{6.1}$$

defines the Feynman rules as

$$\underset{\phi_{k,\omega}}{\bullet} \xrightarrow{\quad G^\phi(k,\omega) \quad} \underset{\phi_{-k,-\omega}}{\bullet} \;, \qquad \overset{J_x}{\underset{\frac{1}{2\sqrt{N}}}{\phi_{x,\tau} \bullet \cdots \bullet \phi_{x,\tau}}} \;. \tag{6.2}$$

As $N \to \infty$, the disorder doesn't affect the system since it couples with $\phi^2$ at the order of $\frac{1}{\sqrt{N}}$. At the order of $1/N$, the disorder averaged 2-point function of $\phi$ is corrected to be

$$G_\phi^B(k,\omega) = \overline{\langle \phi_{k,\omega} \phi_{-k,-\omega} \rangle_J^c} = \underset{\bullet \quad \bullet}{G_\phi(k,\omega)} + \underset{\bullet \quad L_9 \quad \bullet}{\overset{\cdots}{\phantom{x}}}$$

$$= \frac{1}{k^2 + \omega^2} - \frac{W^2}{(4\pi)^2 N} \frac{2\mu^{-\epsilon}}{\epsilon} \frac{\omega^2}{(k^2 + \omega^2)^2}. \tag{6.3}$$

In terms of the renormalized variables defined as before

$$\phi_R = Z_\phi^{-1} \phi, \quad \phi_R^2 = Z_{\phi^2}^{-1} \phi^2, \quad \tau = Z_\tau \tau_R, \quad \omega = \omega_R / Z_\tau, \quad Z_W W_R = W, \tag{6.4}$$

we can then get the finite renormalized correlation function $G_\phi^R = Z_\tau^{-1} Z_\phi^{-2} G_\phi^B$. The divergence in the bare correlation function fixes the renormalization constants to be

$$Z_\tau = 1 + \frac{W_R^2}{16\pi^2 N} \frac{\mu^{-\epsilon}}{\epsilon},$$

$$Z_\phi = 1 - \frac{W_R^2}{32\pi^2 N} \frac{\mu^{-\epsilon}}{\epsilon}. \tag{6.5}$$

As a result, the dynamical exponent and the anomalous dimension of $\phi$ are

$$z = 1 - \mu \partial_\mu \ln Z_\tau = 1 + \frac{W_R^2}{16\pi^2 N}, \tag{6.6}$$

$$\gamma_\phi = \mu \partial_\mu \ln Z_\phi = \frac{W_R^2}{32\pi^2 N}, \tag{6.7}$$

respectively. We can also compute the bare disorder averaged disconnected 4-point function of $\phi$ defined as

$$G_\phi^{B(4pt)} = \Gamma_{4\phi}^B G_\phi^B \left( \frac{k}{2} + q, \frac{\omega}{2} + \omega' \right) G_\phi^B \left( \frac{k}{2} + p, \frac{\omega}{2} + \omega'' \right) G_\phi^B \left( \frac{k}{2} - q, \frac{\omega}{2} - \omega' \right) G_\phi^B \left( \frac{k}{2} - p, \frac{\omega}{2} - \omega'' \right) \tag{6.8}$$

where $\Gamma_{4\phi}^B$ is the 4-point vertex. Then, the renormalized 4-point function is given by

$$G_\phi^{R(4pt)} = \Gamma_{4\phi}^R G_\phi^R \left(\frac{k}{2} + q, \frac{\omega_R}{2} + \omega'\right) G_\phi^R \left(\frac{k}{2} + p, \frac{\omega_R}{2} + \omega''\right) G_\phi^R \left(\frac{k}{2} - q, \frac{\omega_R}{2} - \omega'\right) G_\phi^R \left(\frac{k}{2} - p, \frac{\omega_R}{2} - \omega''\right)$$

$$= Z_\tau^{-4} Z_\phi^{-8} \Gamma_{4\phi}^R G_\phi^B \left(\frac{k}{2} + q, \frac{\omega_R}{2} + \omega'\right) G_\phi^B \left(\frac{k}{2} + p, \frac{\omega_R}{2} + \omega''\right) G_\phi^B \left(\frac{k}{2} - q, \frac{\omega_R}{2} - \omega'\right) G_\phi^B \left(\frac{k}{2} - p, \frac{\omega_R}{2} - \omega''\right)$$

$$= Z_\tau^{-2} Z_\phi^{-4} G_\phi^{B(4pt)}, \tag{6.9}$$

which relates the renormalized and the bare 4-point vertex in the following way

$$\Gamma_{4\phi}^R = Z_\tau^2 Z_\phi^4 \Gamma_{4\phi}^B. \tag{6.10}$$

At the tree level, the bare 4-point vertex is $W^2/N$. The $1/N$ correction is contributed by the 1PI diagram at the order of $1/N^2$. To this end, the bare vertex is given by

$$= \frac{W^2}{N} + \frac{2W^4}{N} \int \frac{d^d p}{(2\pi)^d} \frac{1}{(p^2 + \omega_1^2)[(k_3 - k_1 + p)^2 + \omega_1^2]}$$

$$= \frac{W^2}{N} \left[1 + \frac{W^2}{4\pi^2 N} \frac{\mu^{-\epsilon}}{\epsilon}\right]. \tag{6.11}$$

This determines the renormalization constant to be $Z_W = 1 - \frac{W_R^2}{8\pi^2 N} \frac{\mu^{-\epsilon}}{\epsilon}$, and the $\beta$ function of the randomness is

$$\beta_{W^2} = -\frac{dW_R^2}{d\ln\mu} = \epsilon W_R^2 + W_R^2 \frac{d\ln Z_W^2}{d\ln\mu} = \epsilon W_R^2 + \frac{W_R^4}{4\pi^2 N}. \tag{6.12}$$

At positive $\epsilon$, there is an unphysical attractive fixed point at $\frac{W_R^2}{N} = -4\pi^2\epsilon$. While at negative $\epsilon$, we can find a physical but unstable fixed point at $\frac{W_R^2}{N} = -4\pi^2\epsilon$. This fixed point has two relevant directions including both the disorder and the mass perturbation.

Besides, according to the relation in Eq. (2.14), one can get the renormalization constant associated with the composite operator $\phi^2$ as

$$Z_{\phi^2} = Z_\tau^{-1} Z_W^{-1} = 1 + \frac{W_R^2}{16\pi^2 N} \frac{\mu^{-\epsilon}}{\epsilon}, \tag{6.13}$$

which finds its anomalous dimension to be

$$\gamma_{\phi^2} = \mu\partial_\mu \ln Z_{\phi^2} = -\frac{W_R^2}{16\pi^2 N}. \tag{6.14}$$

Compared to $2\gamma_\phi$, it has an opposite sign. This indicates that the scaling dimension of the singlet operator decreases as $W_R^2$ becomes larger. It is possible that the interaction term $(\phi^2)^2$ becomes relevant if the randomness is strong enough. This would induce extra RG flow towards strong interaction regime above three spatial dimensions. This result is checked by directly computing the renormalization of the $\phi^2$ field, as studied in Appendix F.

## B. Momentum Shell RG

We can alternatively use the momentum shell RG to find how the randomness $W^2$ flows under coarse graining. Let us begin with the action in Eq. (6.1) again. The bosonic field is scattered by the quenched disorder. This process only transfers the momentum, which mixes the fast and slow modes. As the integration of the frequency shell doesn't change the form of effective action, we thus focus on how the action gets modified after integrating over the momentum shell. In order to do this, we separate the field $\phi$ into the fast mode and the slow mode, i.e. $\phi = \phi^> + \phi^<$. Then, in the momentum space, we can write done the action as $S = S^< + S^> + S^{\mathrm{mix}}$, where

$$S^< = \frac{1}{2} \int_0^{\Lambda_0} \frac{d\omega}{2\pi} \int_0^{\Lambda e^{-\ell}} \frac{d^d k}{(2\pi)^d} \left[ (k^2 + \omega^2)\phi_{k,\omega}^< \phi_{-k,-\omega}^< + \frac{1}{\sqrt{N}} \int_0^{\Lambda e^{-\ell}} \frac{d^d p}{(2\pi)^d} J_{k-p} \phi_{k,\omega}^< \phi_{-p,-\omega}^< \right] \quad (6.15)$$

$$S^> = \frac{1}{2} \int_0^{\Lambda_0} \frac{d\omega}{2\pi} \int_{\Lambda e^{-\ell}}^{\Lambda} \frac{d^d k}{(2\pi)^d} \left[ (k^2 + \omega^2)\phi_{k,\omega}^> \phi_{-k,-\omega}^> + \frac{1}{\sqrt{N}} \int_{\Lambda e^{-\ell}}^{\Lambda} \frac{d^d p}{(2\pi)^d} J_{k-p} \phi_{k,\omega}^> \phi_{-p,-\omega}^> \right], \quad (6.16)$$

$$S^{\mathrm{mix}} = \frac{1}{\sqrt{N}} \int_0^{\Lambda_0} \frac{d\omega}{2\pi} \int_{\Lambda e^{-\ell}}^{\Lambda} \frac{d^d k}{(2\pi)^d} \int_0^{\Lambda e^{-\ell}} \frac{d^d p}{(2\pi)^d} J_{k-p} \phi_{k,\omega}^> \phi_{-p,-\omega}^<. \quad (6.17)$$

The renormalization scale is defined as $\ell = -\ln\left[1 - \frac{\delta\Lambda}{\Lambda}\right] > 0$ where $\delta\Lambda$ is the decrease in the UV cutoff of momentum. In other words, $\delta\Lambda \approx \ell\Lambda$. So larger $\ell$ results in a longer length scale after the integration over fast modes. The UV cutoff of frequency is chosen to be $\Lambda_0 \ll \Lambda$. The disordered coupling $J_k$ is independent of frequency and is drawn from the Gaussian distribution

$$P[J] = \frac{1}{\mathcal{N}} \exp\left( -\frac{1}{2W^2} \int d^d k \, [J_k - J_0 \delta^d(k)][J_{-k} - J_0 \delta^d(-k)] \right). \quad (6.18)$$

Here, we take a generic form with a nonzero mean value $J_0$, i.e. $\overline{J_k} = J_0 \delta^d(k)$. Higher moments of $J_k$ are functions of both $J_0$ and $W^2$. A few of them are listed in Appendix G. Nonzero $J_0$ can also be viewed as a mass term in the UV action. To make it clear, we can define a new disorder coupling $J'_k = J_k - J_0 \delta^d(k)$. Then, $J'_k$ obeys the Gaussian distribution with zero mean while there exists an additional mass term $\frac{1}{2\sqrt{N}} J_0 \int \frac{d^d k}{(2\pi)^d} \frac{d\omega}{2\pi} \phi_{k,\omega} \phi_{-k,-\omega}$ in the action. We can define the Feynman rules based on the action $S$.

$$\phi_{k,\omega}^< \overset{G^<(k,\omega)}{\bullet\!\!-\!\!\bullet} \phi_{-k,-\omega}^< \;, \qquad \phi_{k,\omega}^> \overset{G^>(k,\omega)}{\bullet\!-\!-\!\bullet} \phi_{-k,-\omega}^> \;, \qquad \phi_{k,\omega}^{<(>)} \overset{J_{k-p}}{\underset{\frac{1}{2\sqrt{N}}}{\bullet\!\!-\!\!\bullet}} \phi_{-p,-\omega}^{<(>)} \;, \qquad \phi_{k,\omega}^> \overset{J_{k-p}}{\underset{\frac{1}{\sqrt{N}}}{\bullet\!-\!-\!\bullet}} \phi_{-p,-\omega}^< \;. \quad (6.19)$$

After integrating out the fast modes, the effective action at the order of $\frac{1}{N}$ becomes

$$S^< = \frac{1}{2} \int_0^{\Lambda_0} \frac{d\omega}{2\pi} \int_0^{\Lambda e^{-\ell}} \frac{d^d k}{(2\pi)^d} \left[ (k^2 + \omega^2)\phi_{k,\omega}^< \phi_{-k,-\omega}^< + \frac{1}{\sqrt{N}} \int_0^{\Lambda e^{-\ell}} \frac{d^d p}{(2\pi)^d} \, \mathcal{J}_{k,p}(\omega) \, \phi_{k,\omega}^< \phi_{-p,-\omega}^< \right] \quad (6.20)$$

where $p$ and $k$ are both smaller than $\Lambda e^{-\ell}$. The second term is represented as a summation of diagrams

$$
\underbrace{\overset{\displaystyle J_{k-p}}{\underset{\phi^<_{k,\omega}\,\,\,\,\frac{1}{2\sqrt{N}}\,\,\,\,\phi^<_{-p,-\omega}}{\bullet\!\!-\!\!-\!\!\bullet}}}_{} \;-\; \underbrace{\overset{\displaystyle J_{k-q}\quad J_{q-p}}{\underset{\phi^<_{k,\omega}\,\,\,\,G^>(q,\omega)\,\,\,\,\phi^<_{-p,-\omega}}{\bullet\cdots\bullet}}}_{} \;+\; \underbrace{\overset{\displaystyle J_{k-q_1}\;J_{q_1-q_2}\;J_{q_2-p}}{\underset{\phi^<_{k,\omega}\,\,\,q_1\,\,\,q_2\,\,\,\phi^<_{-p,-\omega}}{\bullet\cdots\bullet\cdots\bullet}}}_{} \;+\; \mathcal{O}(1/N^2). \quad (6.21)
$$

As a result, the effective coupling of the singlet operator can be expanded as a power series of $J_k$:

$$
\mathcal{J}_{k,p}(\omega) = J_{k-p} - \frac{1}{\sqrt{N}} \int_{\Lambda e^{-\ell}}^{\Lambda} \frac{d^d q}{(2\pi)^d} J_{k-q} J_{q-p} G^>(q,\omega)
$$
$$
+ \frac{1}{N} \int_{\Lambda e^{-\ell}}^{\Lambda} \frac{d^d q_1 d^d q_2}{(2\pi)^{2d}} J_{k-q_1} J_{q_1-q_2} J_{q_2-p} G^>(q_1,\omega) G^>(q_2,\omega) + \ldots , \quad (6.22)
$$

which is in general a function of frequency $\omega$. Now the distribution of $\mathcal{J}_{k,p}$ is different from the original Gaussian distribution $P[J]$ in Eq. (6.18). Up to the order of $\frac{1}{N}$, the mean and variance are corrected to be

$$
\overline{\mathcal{J}_{k,p}(\omega)} = \overline{J_{k-p}} - \frac{1}{\sqrt{N}} \int_{\Lambda e^{-\ell}}^{\Lambda} \frac{d^d q}{(2\pi)^d} \overline{J_{k-q} J_{q-p}} G^>(q,\omega)
$$
$$
+ \frac{1}{N} \int_{\Lambda e^{-\ell}}^{\Lambda} \frac{d^d q_1 d^d q_2}{(2\pi)^{2d}} \overline{J_{k-q_1} J_{q_1-q_2} J_{q_2-p}} G^>(q_1,\omega) G^>(q_2,\omega) + \mathcal{O}(1/N^{\frac{3}{2}}),
$$
$$
\overline{\mathcal{J}_{k_1,p_1} \mathcal{J}_{k_2,p_2}} - \overline{\mathcal{J}_{p_1,k_1}}\,\overline{\mathcal{J}_{p_2,k_2}} = \overline{J_{k_1-p_1} J_{k_2-p_2}} + \frac{2}{N} \int_{\Lambda e^{-\ell}}^{\Lambda} \frac{d^d q_1 d^d q_2}{(2\pi)^{2d}}
$$
$$
\times \left( \overline{J_{k_1-p_1} J_{k_2-q_1} J_{q_1-q_2} J_{q_2-p_2}} - \overline{J_{k_1-p_1}}\,\overline{J_{k_2-q_1} J_{q_1-q_2} J_{q_2-p_2}} \right)
$$
$$
\times G^>(q_1,\omega) G^>(q_2,\omega) + \mathcal{O}(1/N^{\frac{3}{2}}). \quad (6.23)
$$

Provided that the internal momenta $q$ and $q_{1,2}$ within the momentum shell are much larger than the external momenta in the long wavelength limit, the distribution of disorder is parametrized by the following mean and variance

$$
\overline{\mathcal{J}_{k,p}(\omega)} = \left[ J_0 - \frac{W^2}{\sqrt{N}} \eta_1(\omega) + \frac{W^2 J_0}{N} \eta_2(\omega) \right] \delta^d(k-p) + \ldots \quad (6.24)
$$
$$
\overline{\mathcal{J}_{k_1,p_1}(\omega) \mathcal{J}_{k_2,p_2}(\omega)} - \overline{\mathcal{J}_{k_1,p_1}(\omega)}\,\overline{\mathcal{J}_{k_2,p_2}(\omega)} = \left[ W^2 + \frac{2W^4}{N} \eta_2(\omega) \right] \delta^d(k_1+k_2-p_1-p_2), \quad (6.25)
$$

where up to the first order in $\ell$ and $\omega^2$, we have the approximations

$$
\eta_1(\omega) = \int_{\Lambda e^{-\ell}}^{\Lambda} \frac{d^d q}{(2\pi)^d} G^>(q,\omega) \approx \frac{\ell}{8\pi^2}(\Lambda^2 - \omega^2),
$$
$$
\eta_2(\omega) = \int_{\Lambda e^{-\ell}}^{\Lambda} \frac{d^d q}{(2\pi)^d} [G^>(q,\omega)]^2 \approx \frac{\ell}{8\pi^2}\left(1 - \frac{2\omega^2}{\Lambda^2}\right). \quad (6.26)
$$

As $\Lambda^2 \gg \omega^2$, the second term in $\eta_2(\omega)$ can be ignored. Notice that in Eq. (6.24), the mean value of the random coupling is shifted by a function of $\omega$. In the limit of low frequency, we are able

to eliminate this $\omega$ dependence by redefining the frequency. This leads to a new disorder coupling $\mathcal{J}'_{k,p} = \mathcal{J}_{k,p}(\omega) - \frac{W^2\ell}{8\pi^2\sqrt{N}}\omega^2$, whose mean is independent of $\omega$. Consequently, after integrating a frequency shell with thickness $z\delta\Lambda$, the effective action in Eq. (6.20) is modified to

$$S_{\text{eff}} = \frac{1}{2}\int_0^{\Lambda_0 e^{-z\ell}}\frac{d\omega}{2\pi}\int_0^{\Lambda e^{-\ell}}\frac{d^d k}{(2\pi)^d}\left[(k^2+\omega_R^2)\phi^<_{k,\omega}\phi^<_{-k,-\omega} + \frac{1}{\sqrt{N}}\int_0^{\Lambda e^{-\ell}}\frac{d^d p}{(2\pi)^d}\ \mathcal{J}'_{k,p}\ \phi^<_{k,\omega}\phi^<_{-p,-\omega}\right] \quad (6.27)$$

Up to the order of $\frac{1}{N}$, we have

$$\omega_R = \omega\exp[(z-1)\ell], \quad \text{and} \quad z = 1 + \frac{W^2}{16\pi^2 N}. \quad (6.28)$$

The quantity $z$ is known as the dynamical exponent, consistent with what we got in Eq. (6.6) using field theoretical RG. Then, we want to restore the UV cutoffs by doing rescaling with $\omega \to \omega_R e^{-z\ell}$ and $k \to ke^{-l}$. This results in an action

$$S_{\text{eff}} = \frac{1}{2}\int_0^{\Lambda_0}\frac{d\omega_R}{2\pi}\int_0^{\Lambda}\frac{d^d k}{(2\pi)^d}\left[(k^2+\omega_R^2)\phi^<_{R;k,\omega}\phi^<_{R;-k,-\omega}\right.$$
$$\left. + \frac{1}{\sqrt{N}}\int_0^{\Lambda}\frac{d^d p}{(2\pi)^d}\ e^{(2-d)\ell}\mathcal{J}'_{ke^{-\ell},pe^{-\ell}}\ \phi^<_{R;k,\omega}\phi^<_{R;-p,-\omega}\right]. \quad (6.29)$$

We can find the first term returns to the same form as Eq. (6.1) if the renormalized fundamental field $\phi$ is defined as as $\phi_{R;k,\omega} = e^{-\frac{(d+z+2)}{2}\ell}\phi_{ke^{-\ell},\omega e^{-z\ell}}$. This means the field $\phi$ acquires an anomalous dimension $\gamma_\phi = \frac{W^2}{32\pi^2 N}$. In the second term, the UV action can be reproduced if the renormalized disorder coupling is defined as

$$\mathcal{J}'_{R;k,p} = e^{(2-d)\ell}\mathcal{J}'_{ke^{-\ell},pe^{-\ell}}. \quad (6.30)$$

Then, the RG flow of the mean value of the disorder coupling $\overline{\mathcal{J}'_{k,p}}$ is described by the $\beta$ function

$$\beta_J = \frac{d\mathcal{J}_0}{d\ell} = 2\mathcal{J}_0 - \frac{W^2}{8\pi^2\sqrt{N}} + \frac{W^2\mathcal{J}_0}{8\pi^2 N}, \quad (6.31)$$

where $\mathcal{J}_0 = J_0/\Lambda^2$. Notice it can flow to a nonzero value even though it is set to be zero in the UV. However, no matter what value it is, we can always fine tune a constant UV mass to cancel it such that the system remains critical. Besides, one can also define the renormalized width of the Gaussian distribution as

$$W_R^2 = W^2\exp\left((4-d)\ell + \frac{W^4}{4\pi^2 N}\ell\right) \quad (6.32)$$

It has the $\beta$ function

$$\beta_{W^2} = \frac{dW_R^2}{d\ell} = \epsilon W_R^2 + \frac{W_R^4}{4\pi^2 N}, \quad (6.33)$$

at $d = 4 - \epsilon$. This equation is the same as Eq. (6.12). As we described before, the width of the Gaussian distribution keeps increasing as we go to a larger length scale below $d = 4$ spatial dimensions. At the order of $1/N$, there is no sign of any physical fixed point. We can compute higher order corrections of these $\beta$ functions. But still, no fixed point can be reliably obtained in this perturbative calculation up to $1/N^2$ order, as we discussed in Appendix G.

## VII. DISCUSSION AND CONCLUSION

In this work, we have studied the weak disorder effect on the clean Wilson-Fisher fixed point and the free Gaussian fixed point. The RG flow driven by the random mass disorder is encoded in the $\beta$-functions given by Eq. (5.13) at $d = 2 + \epsilon$ and given by Eq. (6.12) or, equivalently, Eq. (6.33) at $d = 4 - \epsilon$.

At $2 + \epsilon$ and finite $N$, there exists an attractive fixed point at $\frac{W_R^2}{N} = \frac{\pi}{256}(\epsilon + \frac{64}{3\pi^2 N})$. At $\epsilon = 0$, this is the disordered fixed point at $d = 2$ found in previous work [6]. Below two spatial dimensions, there is still an attractive fixed point if $N|\epsilon| < \frac{64}{3\pi^2}$. This is because the $1/N$ correction reduces the lower critical dimension. Concretely, at the order of $1/N$, the anomalous dimension of the $\sigma$ operator given by Eq. (5.10) is negative provided $W_R^2 = 0$ in the clean system. Thus, the lower critical spatial dimension is corrected to be $d' = 2 - 2\gamma_\sigma|_{W_R^2 = 0} < 2$. Notice as $N \to \infty$, the universal data of this fixed point at finite $\epsilon$, such as the anomalous dimension $\gamma_\sigma^*$, cannot be continuously deformed from what we listed in Tab. I. The reason for this discontinuity is that the limit of $N \to \infty$ and the disorder averaged scaling limit do not commute at $\epsilon > 0$. In order to get the physical behavior, we should take the IR limit first. This requires the double expansion of $\epsilon$ and $1/N$, which cannot be obtained by a $1/N$ expansion at finite $\epsilon$. The system at $N = \infty$ is of pure theoretical interest.

At $4 - \epsilon$, the system is decoupled with disorder at $N = \infty$. At finite but large $N$, disorder will drive the system away from the Gaussian fixed point below four spatial dimensions. At positive $\epsilon$, there is a stable fixed point at negative randomness which can never be reached in a physical system. This concrete RG calculation actually reproduce the instability encoded in the replicated action being not bounded from below. If $\epsilon$ is negative, there can be a physical fixed point at positive $\frac{W_R^2}{N}$. However, it is unstable. This means, even though weak disorder is irrelevant at the Gaussian fixed point, a strong enough mass disorder will still be significantly enhanced in the IR.

There is one interesting possibility that the stable fixed point at $2 + \epsilon$ survives at $d = 3$ at finite interaction strength. If this is the case, we expect its universal behavior already encoded in the critical exponents in Eq. (5.14) and Eq. (5.15) as analytical functions of $\epsilon$ and $1/N$. At $N = 2$ and $\epsilon = 1$, we can get $z^* = \frac{3}{2} + \frac{16}{3\pi^2} \approx 2.04$ and $\Delta_\sigma \approx 3.54$ which leads to $\nu \approx \frac{2}{3} = \frac{2}{d}$ at $d = 3$. This saturates the bound given by the Harris criterion, i.e. $\nu d = 2$, which doesn't have to be true if we include higher order corrections. This fixed point should describe the phase transition between a glassy phase and the superfluid phase in a $3 + 1$ dimensional interacting bosonic system perturbed by off-diagonal disorder. In the presence of the particle-hole symmetry, the glassy phase is expected to be an incompressible Mott glass[23]. Thus, we suggest that this phase transition is continuous. Hopefully the critical exponents predicted above can be testified in the future numerical and experimental studies. It is possible that more fixed points exist at

larger randomness which renders the situation more complicated. Sophisticated non-perturbative technique is needed to study the strong disorder regime.

In conclusion, we study the random mass O(N) vector model at $2 < d < 4$. At $N = \infty$, the disordered theory can be solved exactly. We have two ways to understand its scale invariance in the IR. Around the fixed point at $N = \infty$ and $d = 2$ as well as the fixed point at $N = \infty$ and $d = 4$, we can perturbatively obtain the RG flow around the Wilson-Fisher fixed point and the Gaussian fixed point, respectively at $d = 2 + \epsilon$ and $4 - \epsilon$, by using the double expansion of $\epsilon$ and $1/N$. The results are shown in Fig. 1. It helps us to postulate the fixed point structure at $d = 3$, where the system has experimental realizations. By all means, a concrete study of the $d = 3$ system calls for non-perturbative methods. We also hope our study can stimulate more numerical or experimental study on the bosonic system with random mass disorder, especially at $d = 3$.

**ACKNOWLEDGEMENT**

HM specially thank Sung-Sik Lee for discussions and collaboration on related subjects. HM would also like to thank Chong Wang, Hart Goldman, Alex Thomason, Zhen Bi for inspiring discussion. Research at Perimeter Institute is supported in part by the Government of Canada through the Department of Innovation, Science and Economic Development Canada and by the Province of Ontario through the Ministry of Colleges and Universities.

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

# Appendices

At the clean Wilson-Fisher fixed point, the system is described by the critical O(N) model in Eq. (3.3) with $J_x = 0$. The $\phi$ field is only coupled to $\sigma$ at order of $\frac{1}{N}$. So in the large N limit, the propagator of $\phi$ is the same as in the free theory, i.e.

$$G_\phi(K) = \langle \phi_K \phi_{-K} \rangle = \quad \bullet\!\!\!-\!\!\!-\!\!\!-\!\!\!\bullet \quad = K^{-2}. \tag{A.1}$$

Here $K$ is a short notation for $(k, \omega)$. Since the propagation of $\sigma$ is driven by its interaction with $\phi$, it is thus of order $1/N$ and is given by

$$\langle \sigma_K \sigma_{-K} \rangle = \underset{G_\sigma(K)}{\sim\!\!\!\sim\!\!\!\sim\!\!\!\sim} = \frac{2\lambda}{N}\bullet\!\!\!-\!\!\!-\!\!\!\bullet + \cdots \overset{G_\phi}{\bigcirc} \cdots + \cdots \overset{G_\phi}{\bigcirc}\!-\!\bigcirc \cdots + \ldots$$

$$= \frac{2\lambda}{N} + \left(\frac{2\lambda}{N}\right)^2 \frac{N}{2} \int \frac{d^{d+1}P}{(2\pi)^{d+1}} \frac{1}{P^2(K-P)^2} + \left(\frac{2\lambda}{N}\right)^3 \left[\frac{N}{2} \int \frac{d^{d+1}P}{(2\pi)^{d+1}} \frac{1}{P^2(K-P)^2}\right]^2$$

$$+ \ldots. \tag{A.2}$$

The integration in each loop can be evaluated as $I = \int \frac{d^{d+1}P}{(2\pi)^{d+1}} \frac{1}{P^2(K-P)^2} = c_2 K^{d-3}$ where $c_2 = -\frac{2^{-(d+1)}(4\pi)^{\frac{2-d}{2}}}{\Gamma(\frac{d}{2})\sin\frac{\pi(d+1)}{2}}$. It is positive at $2 \leq d < 3$. Then, the sum of all the diagrams gives $\langle \sigma_K \sigma_{-K} \rangle = \frac{2\lambda}{N} \frac{1}{1+c_2\lambda K^{d-3}}$. When $d < 3$, as $K \to 0$ in the low energy limit, $\langle \sigma_K \sigma_{-K} \rangle \approx \frac{2}{c_2 N} K^{3-d}$ which is independent of UV coupling $\lambda$. As we make the change $\sigma \to \frac{1}{\sqrt{N}}\sigma$ in the main context, we can rewrite the propagator as $G_\sigma(K) = \frac{2}{c_2} K^{3-d}$.

## Appendix B: Exact study of the model in Eq.(4.1)

In this sections, we compute the correlation functions of the theory in Eq.(4.1) as functions of the disordered coupling. Given the action in Eq.(4.1), we can add a source $t_{x,\tau}$ coupled to the generalized free field (GFF) $\sigma_{x,\tau}$. Then, the total action in the real space is

$$S = \frac{1}{2} \int d^d x d\tau d^d x' d\tau' \; \sigma_{x,\tau} G_\sigma^{-1}(|x-x'|, |\tau-\tau'|)\sigma_{x',\tau'} + \int d^d x d\tau \; (iJ_x + t_{x,\tau})\sigma_{x,\tau}. \tag{B.1}$$

It is easy to compute the partition functional $Z[J,t]$ by integrating over $\sigma$, which is a functional of disordered coupling $J$ and source $t$.

$$Z[J,t] = Z_0 \exp\left\{\frac{1}{2} \int d^d x d\tau d^d x' d\tau'(iJ_x + t_{x,\tau})G_\sigma(|x-x'|, |\tau-\tau'|)(iJ_{x'} + t_{x',\tau'})\right\}, \tag{B.2}$$

where $Z_0 = \sqrt{\frac{(2\pi)^V}{\det G_\sigma^{-1}}}$ is a constant depending on systems size $V$. Then, the correlation functions of $\sigma_{x,\tau}$ can be obtained by taking derivative of $Z$ with respect of $t$ followed by setting $t = 0$. Here we list the exact results of a few correlation functions.

$$\langle\sigma_{x_0,\tau_0}\rangle_J = -\frac{1}{Z[J,t]}\frac{\partial Z[J,t]}{\partial t_{x_0,\tau_0}}\Big|_{t=0} = -i\int d^dx d\tau J_x G_\sigma(|x-x_0|,|\tau-\tau_0|),$$

$$\langle\sigma^2_{x_0,\tau_0}\rangle_J = \frac{1}{Z[J,t]}\frac{\partial^2 Z[J,t]}{\partial t^2_{x_0,\tau_0}}\Big|_{t=0}$$

$$= -\int d^dx_1 d\tau_1 d^dx_2 d\tau_2 J_{x_1} J_{x_2} G_\sigma(|x_1-x_0|,|\tau_1-\tau_0|)G_\sigma(|x_2-x_0|,|\tau_2-\tau_0|),$$

$$\langle\sigma_{x_1,\tau_1}\sigma_{x_2,\tau_2}\rangle_J = \frac{1}{Z[J,t]}\frac{\partial}{\partial t_{x_2,\tau_2}}\frac{\partial}{\partial t_{x_1,\tau_1}}Z[J,t]\Big|_{t=0} = G_\sigma(x_{12},\tau_{12})$$

$$- \int d^dx_3 d\tau_3 d^dx_4 d\tau_4 J_{x_3} J_{x_4} G_\sigma(x_{13},\tau_{13})G_\sigma(x_{24},\tau_{24}),$$

$$\langle\sigma_{x_1,\tau_1}\sigma_{x_2,\tau_2}\rangle^c_J = \frac{\partial}{\partial t_{x_2,\tau_2}}\frac{\partial}{\partial t_{x_1,\tau_1}}\ln Z[J,t]\Big|_{t=0} = G_\sigma(x_{12},\tau_{12}),$$

$$\langle\sigma^2_{x_1,\tau_1}\sigma^2_{x_2,\tau_2}\rangle_J = \frac{1}{Z[J,t]}\frac{\partial^2}{\partial t^2_{x_2,\tau_2}}\frac{\partial^2}{\partial t^2_{x_1,\tau_1}}Z[J,t]\Big|_{t=0} = 2[G_\sigma(x_{12},\tau_{12})]^2$$

$$+ \int d^dx_3 d\tau_3 d^dx_4 d\tau_4 d^dx_5 d\tau_5 d^dx_6 d\tau_6 J_{x_3}J_{x_4}J_{x_5}J_{x_6}G_\sigma(x_{13},\tau_{13})G_\sigma(x_{14},\tau_{14})G_\sigma(x_{25},\tau_{25})G_\sigma(x_{26},\tau_{26})$$

$$- 4G_\sigma(x_{12},\tau_{12})\int d^dx_3 d\tau_3 d^dx_4 d\tau_4 J_{x_3}J_{x_4}G_\sigma(x_{13},\tau_{13})G_\sigma(x_{24},\tau_{24}),$$

$$\langle\sigma^2_{x_1,\tau_1}\sigma^2_{x_2,\tau_2}\rangle^c_J = \langle\sigma^2_{x_1,\tau_1}\sigma^2_{x_2,\tau_2}\rangle_J - \langle\sigma^2_{x_1,\tau_1}\rangle\langle\sigma^2_{x_2,\tau_2}\rangle_J$$

$$= 2[G_\sigma(x_{12}\tau_{12})]^2 - 4G_\sigma(x_{12},\tau_{12})\int d^dx_3 d\tau_3 d^dx_4 d\tau_4 J_{x_3}J_{x_4}G_\sigma(x_{13},\tau_{13})G_\sigma(x_{24},\tau_{24}). \tag{B.3}$$

We use the same notation as in the main text where the subscript "$J$" denotes the disorder coupling dependence and the superscript "$c$" labels the connected correlation functions. These expressions can be represented by Feynman diagrams given in Eq. (4.2), Eq. (4.3) and Eq. (4.4). More connected and disconnected correlation functions can be computed in the same way.

### Appendix C: Replica trick

In this section, we use the replica trick to study the theory of GFF in Eq. (4.1). Based on the equality $\ln Z[J] = \lim_{n\to 0}\frac{Z^n[J]-1}{n}$, we can sum over $n$ copies of the original theory and then do the disorder averaging to get a replicated theory. In order to evaluate the correlation functions, we can introduce the source $t$ and $t'$ coupled to $\sigma$ and $\sigma^2$ respectively. The disorder averaged 2-point

functions of $\sigma$ and $\sigma^2$ are naturally given by

$$\overline{\langle \sigma_{k,\omega}\sigma_{-k,-\omega}\rangle^c_J} = (-1)^2 \int \mathcal{D}J \ P[J] \frac{\partial}{\partial t_{k,\omega}} \frac{\partial}{\partial t_{-k,-\omega}} \ \ln Z[J,t,t']\Big|_{t=t'=0}$$

$$= \lim_{n\to 0} \frac{1}{n} \frac{\partial}{\partial t_{k,\omega}} \frac{\partial}{\partial t_{-k,-\omega}} \overline{Z^n}[W^2,t,t']\Big|_{t=t'=0} = \lim_{n\to 0} \frac{1}{n} \sum_{\alpha,\alpha'} \langle \sigma_{\alpha,k,\omega}\sigma_{\alpha',-k,-\omega}\rangle, \quad \text{(C.1)}$$

$$\overline{\langle \sigma^2_{k,\omega}\sigma^2_{-k,-\omega}\rangle^c_J} = \lim_{n\to 0} \frac{1}{n} \frac{\partial}{\partial t'_{k,\omega}} \frac{\partial}{\partial t'_{-k,-\omega}} \overline{Z^n}[W^2,t,t']\Big|_{t=t'=0} = \lim_{n\to 0} \frac{1}{n} \sum_{\alpha,\alpha'} \langle \sigma^2_{\alpha,k,\omega}\sigma^2_{\alpha',-k,-\omega}\rangle, \quad \text{(C.2)}$$

where the replicated action $\overline{Z^n}[W^2,t,t']$ is

$$\overline{Z^n} = \int \mathcal{D}J \ \mathcal{D}\sigma_\alpha \exp\left\{ -\sum_{\alpha=1}^{n} \int d^d k d\omega \left[ \frac{1}{2}\sigma_{\alpha,k,\omega}G^{-1}_\sigma(k,\omega)\sigma_{\alpha,-k,-\omega} + \ iJ_{-k}\delta(\omega)\sigma_{\alpha,k,\omega} \right] \right\}$$

$$\times \exp\left\{ -\frac{1}{2W^2} \int d^d k \ J_k J_{-k} \right\} \exp\left\{ -\int d^d x d\tau \ t_{x,\tau} \sum_{\alpha=1}^{n} \sigma_{\alpha,x,\tau} - \int d^d x d\tau \ t'_{x,\tau} \sum_{\alpha=1}^{n} \sigma^2_{\alpha,x,\tau} \right\}$$

$$= \int \mathcal{D}\sigma_\alpha \exp\left\{ -\left[ \frac{1}{2} \int d^d k d\omega \sum_{\alpha,\alpha'=1}^{n} \sigma_{\alpha,k,\omega} \left( G^{-1}_\sigma(k,\omega)\delta_{\alpha,\alpha'} + W^2\delta(\omega) \right) \sigma_{\alpha',-k,-\omega} \right] \right\}$$

$$\times \exp\left\{ -\int d^d x d\tau \ t_{x,\tau} \sum_{\alpha=1}^{n} \sigma_{\alpha,x,\tau} - \int d^d x d\tau \ t'_{x,\tau} \sum_{\alpha=1}^{n} \sigma^2_{\alpha,x,\tau} \right\}. \quad \text{(C.3)}$$

The last equality in Eq. (C.1) and Eq. (C.2) are obtained by explicitly taking the derivatives of the replicated theory. In Eq. (C.3), $\alpha$ is the replica index. We can define a matrix whose row and column indices are replica indices. Its element is given by $\mathcal{G}^{-1}_{\alpha,\alpha'}(k,\omega) = G^{-1}_\sigma(k,\omega)\delta_{\alpha,\alpha'} + W^2\delta(\omega)$. It is straightforward to find its inverse exactly as $\mathcal{G}_{\alpha,\alpha'}(k,\omega) = G_\sigma(k,\omega)\delta_{\alpha,\alpha'} - \frac{W^2\delta(\omega)[G_\sigma(k,\omega)]^2}{1+nW^2\delta(\omega)G_\sigma(k,\omega)}$. Recall that $G_\sigma(k,\omega) \propto (k^2 + \omega^2)^{\frac{2\Delta_\sigma-d-1}{2}}$. If we perform the scale transformation by doing the replacements $k \to ke^{-\ell}$ and $\omega \to \omega e^{-\ell}$, where $\ell$ is a change in logarithmic length scale, then each matrix element $\mathcal{G}_{\alpha,\alpha'}$ becomes $G_\sigma(k,\omega)e^{(d+1-2\Delta_\sigma)\ell}\delta_{\alpha,\alpha'} - \frac{W^2\delta(\omega)e^{(2d+3-4\Delta_\sigma)\ell}[G_\sigma(k,\omega)]^2}{1+nW^2\delta(\omega)G_\sigma(k,\omega)e^{(d+2-2\Delta_\sigma)\ell}}$. If $d+2-2\Delta < 0$, then the order of two limits $n \to 0$ and $\ell \to \infty$ commute. Taking these two limits successively, the second term contributing $\mathcal{G}_{\alpha,\alpha'}$ vanishes. As a result, each replica decouples and the disorder effect is irrelevant. While when $d + 2 - 2\Delta > 0$, the two limits don't commute. We need to take $n \to 0$ first and then $\ell \to \infty$ in order to get physical results.

Next, we evaluate the correlation functions in Eq. (C.1) and Eq. (C.2). For $\sigma$, its disorder averaged 2-point function is

$$\overline{\langle \sigma_{k,\omega}\sigma_{-k,-\omega}\rangle^c_J} = \lim_{n\to 0} \left[ G_\sigma(k,\omega) - \frac{nW^2\delta(\omega)[G_\sigma(k,\omega)]^2}{1 + nW^2\delta(\omega)G_\sigma(k,\omega)} \right] = G_\sigma(k,\omega). \quad \text{(C.4)}$$

For the composite operator, we can get

$$
\overline{\langle \sigma_{k,\omega}^2 \sigma_{-k,-\omega}^2 \rangle_J^c} = \lim_{n\to 0} \frac{2}{n} \sum_{\alpha,\alpha'=1}^{n} \int \frac{d^d p\, d\Omega}{(2\pi)^{d+1}} \langle \sigma_\alpha(k-p,\omega-\Omega)\sigma_{\alpha'}(-k+p,-\omega+\Omega)\rangle \langle \sigma_\alpha(p,\Omega)\sigma_{\alpha'}(-p,-\Omega)\rangle
$$

$$
= \lim_{n\to 0} \frac{2}{n} \sum_{\alpha,\alpha'=1}^{n} \int \frac{d^d p\, d\Omega}{(2\pi)^{d+1}} \Big( G_\sigma(k-p,\omega-\Omega)\delta_{\alpha\alpha'} - \frac{W^2\delta(\omega-\Omega)[G_\sigma(k-p,\omega-\Omega)]^2}{1+nW^2\delta(\omega-\Omega)G_\sigma(k-p,\omega-\Omega)} \Big)
$$

$$
\times \Big( G_\sigma(p,\Omega)\delta_{\alpha\alpha'} - \frac{W^2\delta(\Omega)[G_\sigma(p,\Omega)]^2}{1+nW^2\delta(\Omega)G_\sigma(p,\Omega)} \Big)
$$

$$
= 2 \int \frac{d^d p\, d\Omega}{(2\pi)^{d+1}} G_\sigma(k-p,\omega-\Omega)G_\sigma(p,\Omega)
$$

$$
- 4 \lim_{n\to 0} \int \frac{d^d p\, d\Omega}{(2\pi)^{d+1}} \frac{W^2\delta(\Omega)G_\sigma(k-p,\omega-\Omega)[G_\sigma(p,\Omega)]^2}{1+nW^2\delta(\Omega)G_\sigma(p,\Omega)}
$$

$$
+ 2 \lim_{n\to 0} n \int \frac{d^d p\, d\Omega}{(2\pi)^{d+1}} \frac{W^2\delta(\omega-\Omega)[G_\sigma(k-p,\omega-\Omega)]^2}{1+nW^2\delta(\omega-\Omega)G_\sigma(k-p,\omega-\Omega)} \frac{W^2\delta(\Omega)[G_\sigma(p,\Omega)]^2}{1+nW^2\delta(\Omega)G_\sigma(p,\Omega)}. \quad \text{(C.5)}
$$

The physical results by taking the $n \to 0$ limit first are consistent with the exact results listed in Eq. (B.3).

**Appendix D: dimension reduction**

Starting from the replicated theory in Eq. (C.3), we can also derive the supersymmetric effective action discussed in Sec. IV B, as studied in Ref.[40, 41]. Set $t = t' = 0$. We can define $\varphi = \frac{1}{2}(\sigma^1+\rho)$, $\eta = \sigma^1 - \rho$ and $\chi^a = \sigma^a - \rho$ for $a = 2,\ldots,n$, where $\rho = \frac{1}{n-1}(\sigma^2 + \cdots + \sigma^n)$. While $\varphi$ represents the mean value of replica fields, $\eta$ and $\chi^a$ quantify the replica symmetry breaking. In terms of these new fields, the replicated theory can be written in the following way such that in the limit of $n \to 0$, we have $Z^n \to 1$. The disorder averaged $n$-copies of the partition function is given by

$$
\overline{Z^n} = |n-1| \int \mathcal{D}\varphi \mathcal{D}\eta \mathcal{D}\chi^a \; \delta\Big(\sum_{a=2}^{n} \chi^a\Big)
$$

$$
\times \exp\Big\{ -\frac{1}{2} \int d^d k\, d\omega \Big[ (\varphi + \frac{1}{2}\eta)_{k,\omega} G_\sigma^{-1}(k,\omega)(\varphi + \frac{1}{2}\eta)_{-k,-\omega}
$$

$$
+ \sum_{a=2}^{n} (\chi^a + \frac{2\varphi - \eta}{2})_{k,\omega} G_\sigma^{-1}(k,\omega)(\chi^a + \frac{2\varphi - \eta}{2})_{-k,-\omega} \Big]
$$

$$
- \frac{W^2}{2} \int d^d k\, d\omega\, \delta(\omega)[n\varphi + \frac{2-n}{2}\eta]_{k,\omega}[n\varphi + \frac{2-n}{2}\eta]_{-k,-\omega} \Big\}
$$

$$
= |n-1| \int \mathcal{D}\varphi \mathcal{D}\eta \mathcal{D}\chi^a \; \delta\Big(\sum_{a=2}^{n} \chi^a\Big) \exp\Big\{ -S_0 - nS_1 - n^2 S_2 \Big\}, \quad \text{(D.1)}
$$

where the effective action consists

$$S_0 = \int \frac{d^d k d\omega}{(2\pi)^{d+1}} G_\sigma^{-1}(k,\omega)[\eta_{k,\omega}\varphi_{-k,-\omega} + \frac{1}{2}\sum_{a=2}^{n}\chi_{k,\omega}^a\chi_{-k,-\omega}^a] + \frac{W^2}{2}\int \frac{d^d k d\omega}{(2\pi)^{d+1}}\,\delta(\omega)\eta_{k,\omega}\eta_{-k,-\omega},$$

$$S_1 = \frac{1}{2}\int \frac{d^d k d\omega}{(2\pi)^{d+1}} G_\sigma^{-1}(k,\omega)\left[\varphi_{k,\omega}\varphi_{-k,-\omega} - \eta_{k,\omega}\varphi_{-k,-\omega} + \frac{1}{4}\eta_{k,\omega}\eta_{-k,-\omega}\right]$$
$$+ \frac{W^2}{2}\int \frac{d^d k d\omega}{(2\pi)^{d+1}}\,\delta(\omega)(2\varphi_{k,\omega}\eta_{-k,-\omega} - \eta_{k,\omega}\eta_{-k,-\omega}),$$

$$S_2 = \frac{W^2}{2}\int \frac{d^d k d\omega}{(2\pi)^{d+1}}\,\delta(\omega)(\varphi_{k,\omega}\varphi_{-k,-\omega} - \varphi_{k,\omega}\eta_{-k,-\omega} + \frac{1}{4}\eta_{k,\omega}\eta_{-k,-\omega}). \tag{D.2}$$

We can check that $\lim_{n\to 0}\overline{Z^n} = 1$. Therefore, any observable can be expressed as

$$\overline{\langle\mathbf{O}[\sigma]\rangle} = \lim_{n\to 0}\int \mathcal{D}\varphi\mathcal{D}\eta\mathcal{D}\chi^a\,\delta\Big(\sum_{a=2}^{n}\chi^a\Big)\,\mathbf{O}[\varphi,\eta,\chi^a]\,\exp\Big\{-S_0\Big\}. \tag{D.3}$$

Furthermore, in the limit of $n \to 0$, we have equality

$$\lim_{n\to 0}\frac{1}{2}\sum_{a=2}^{n}\chi_{k,\omega}^a G_\sigma^{-1}(k,\omega)\chi_{-k,-\omega}^a = \bar{\psi}_{k,\omega}G_\sigma^{-1}(k,\omega)\psi_{-k,-\omega}, \tag{D.4}$$

where $\psi$ and $\bar{\psi}$ are anti-commuting scalar fields. This can be verified by Gaussian integral over the fields on both sides. Consequently, the effective action becomes

$$\lim_{n\to 0} S_0 = \int \frac{d^d k d\omega}{(2\pi)^{d+1}} G_\sigma^{-1}(k,\omega)[\eta_{k,\omega}\varphi_{-k,-\omega} + \bar{\psi}_{k,\omega}\psi_{-k,-\omega}] + \frac{W^2}{2}\int \frac{d^d k d\omega}{(2\pi)^{d+1}}\,\delta(\omega)\eta_{k,\omega}\eta_{-k,-\omega} \tag{D.5}$$

Notice that the disorder only couples to the zero mode of the $\eta$ field. So we can integrate over all other modes and get the effective action as

$$S_{\text{eff}}^{\omega=0} = \int \frac{d^d k}{(2\pi)^d}\left[\eta_{k,0}G_{k,0}^{-1}\varphi_{-k,0} + \bar{\psi}_{k,0}G_{k,0}^{-1}\psi_{-k,0} + \frac{W^2}{2}\eta_{k,0}\eta_{-k,0}\right]. \tag{D.6}$$

In an isotropic system, by defining a new momentum $p$ in $\mathfrak{D} = \frac{2d}{d+1-2\Delta_\sigma}$ dimensions as what we do in the main text, we can get a free action in terms of new fields $\eta'_p = \eta_{k,0}$, $\varphi'_p = \varphi_{k,0}$, $\psi'_p = \psi_{k,0}$ and $\bar{\psi}'_p = \bar{\psi}_{k,0}$. In the real space, it is

$$\frac{2}{c_2}\mathcal{S}_{\text{eff}}^{\omega=0} = \int d^{\mathfrak{D}}x\left[\eta'_x(-\nabla^2)\varphi'_x + \bar{\psi}'_x(-\nabla^2)\psi'_x + \frac{1}{2}\eta'_x\eta'_x\right], \tag{D.7}$$

where we set $W^2 = \frac{c_2}{2}$. This action is invariant under a transformation

$$\delta\varphi'_x = \bar{a}\epsilon_\mu x_\mu\psi'_x, \quad \delta\eta'_x = 2\bar{a}\epsilon_\mu\partial_\mu\psi'_x, \quad \delta\psi'_x = 0, \quad \delta\bar{\psi}'_x = \bar{a}\epsilon_\mu(-x_\mu\eta'_x + 2\partial_\mu\varphi'_x), \tag{D.8}$$

where $\bar{a}$ is an infinitesimal anticommuting number and $\epsilon_\mu$ is an arbitrary vector. Then, in terms of a superfield $\Phi_x = \varphi'_x + \theta\psi'_x + \bar{\theta}\bar{\psi}'_x - \theta\bar{\theta}\eta'_x$, the effective action in Eq. (D.7) can be written as

$$\mathcal{S}_{\text{eff}}^{\omega=0}[\Phi] = c_2\frac{\Omega_d}{\Omega_{\mathfrak{D}}}\int d^{\mathfrak{D}}x d\theta d\bar{\theta}\left(\Phi_x[-\nabla^2 - 2\partial_{\bar{\theta}}\partial_\theta]\Phi_x\right), \tag{D.9}$$

which is the same as Eq. (4.22) in the main context.

## Appendix E: Loop corrections

In the main context, there are two types of propagators:

$$\underset{(k^2+\omega^2)^{-\alpha}}{\overset{\alpha}{\bullet\!\!-\!\!-\!\!\bullet}} \, , \qquad \underset{(k^2)^{-\alpha}\delta(\omega)}{\overset{\alpha}{\bullet\!\cdots\!\bullet}} \, . \tag{E.1}$$

The one on the left transfers both momentum and energy simultaneously while the other one on the right only transfers momentum. We then compute the four types of subdiagrams below. The results will be used later.

$$\underset{D_1}{\overset{\alpha \quad \beta}{\bullet\!-\!\bullet\!-\!\bullet}} \, , \qquad \underset{D_2}{\overset{\alpha \quad \beta}{\bullet\!-\!\bullet\!\cdots\!\bullet}} \, , \qquad \underset{D_3}{\overset{\alpha}{\diamond}}_{\beta} \, , \qquad \underset{D_4}{\overset{\alpha}{\diamond}}_{\beta} \, . \tag{E.2}$$

The diagrams $D_1 \sim D_4$ are evaluated as following:

$$D_1[\alpha, \beta; k, \omega] = (k^2 + \omega^2)^{-\alpha-\beta}, \tag{E.3}$$

$$D_2[\alpha, \beta; k, \omega] = (k^2)^{-\alpha-\beta}\delta(\omega), \tag{E.4}$$

$$\begin{aligned} D_3[\alpha, \beta; k, \omega] &= \int \frac{d^d p \, d\Omega}{(2\pi)^{d+1}} \frac{1}{(p^2+\Omega^2)^\alpha} \frac{1}{[(k-p)^2+(\omega-\Omega)^2]^\beta} \\ &= \frac{\Gamma[\alpha+\beta-\frac{d+1}{2}]\Gamma[\frac{d+1}{2}-\beta]\Gamma[\frac{d+1}{2}-\alpha]}{(4\pi)^{\frac{d+1}{2}}\Gamma[\alpha]\Gamma[\beta]\Gamma[d+1-\beta-\alpha]} \frac{1}{[k^2+\omega^2]^{\alpha+\beta-\frac{d+1}{2}}}, \end{aligned} \tag{E.5}$$

$$\begin{aligned} D_4[\alpha, \beta; k, \omega] &= \int \frac{d^d p}{(2\pi)^d} \frac{1}{(p^2+\omega^2)^\alpha} \frac{1}{[(k-p)^2]^\beta} \\ &= \frac{\Gamma[\alpha+\beta-\frac{d}{2}]\Gamma[\frac{d}{2}-\beta]}{(4\pi)^{d/2}\Gamma[\alpha]\Gamma[\frac{d}{2}]} {}_2F_1[\frac{d}{2}-\beta, \alpha+\beta-\frac{d}{2}, \frac{d}{2}, \frac{k^2}{k^2+\omega^2}] \frac{1}{(k^2+\omega^2)^{\alpha+\beta-\frac{d}{2}}}. \end{aligned} \tag{E.6}$$

Additionally, we can have following useful integrals.

$$\begin{aligned} D_3'[\alpha, \beta; k, \omega] &= \int \frac{d^d p \, d\Omega}{(2\pi)^{d+1}} \frac{p^2}{(p^2+\Omega^2)^\alpha} \frac{1}{[(k-p)^2+(\omega-\Omega)^2]^\beta} \\ &= \frac{d}{2} \frac{\Gamma[\alpha+\beta-\frac{d+3}{2}]}{(4\pi)^{\frac{d+1}{2}}\Gamma[\alpha]\Gamma[\beta]} \frac{\Gamma[\frac{d+3}{2}-\beta]\Gamma[\frac{d+3}{2}-\alpha]}{\Gamma[d+3-\alpha-\beta]} \frac{1}{[k^2+\omega^2]^{\alpha+\beta-\frac{d+3}{2}}} \\ &\quad + \frac{\Gamma[\alpha+\beta-\frac{d+1}{2}]}{(4\pi)^{\frac{d+1}{2}}\Gamma[\alpha]\Gamma[\beta]} \frac{\Gamma[\frac{d+1}{2}-\beta]\Gamma[\frac{d+5}{2}-\alpha]}{\Gamma[d+3-\alpha-\beta]} \frac{k^2}{[k^2+\omega^2]^{\alpha+\beta-\frac{d+1}{2}}}, \end{aligned} \tag{E.7}$$

$$\begin{aligned} D_3''[\alpha, \beta; k, \omega] &= \int \frac{d^d p \, d\Omega}{(2\pi)^{d+1}} \frac{\Omega^2}{(p^2+\Omega^2)^\alpha} \frac{1}{[(k-p)^2+(\omega-\Omega)^2]^\beta} \\ &= \frac{1}{2} \frac{\Gamma[\alpha+\beta-\frac{d+3}{2}]}{(4\pi)^{\frac{d+1}{2}}\Gamma[\alpha]\Gamma[\beta]} \frac{\Gamma[\frac{d+3}{2}-\beta]\Gamma[\frac{d+3}{2}-\alpha]}{\Gamma[d+3-\alpha-\beta]} \frac{1}{[k^2+\omega^2]^{\alpha+\beta-\frac{d+3}{2}}} \\ &\quad + \frac{\Gamma[\alpha+\beta-\frac{d+1}{2}]}{(4\pi)^{\frac{d+1}{2}}\Gamma[\alpha]\Gamma[\beta]} \frac{\Gamma[\frac{d+1}{2}-\beta]\Gamma[\frac{d+5}{2}-\alpha]}{\Gamma[d+3-\alpha-\beta]} \frac{\omega^2}{[k^2+\omega^2]^{\alpha+\beta-\frac{d+1}{2}}}. \end{aligned} \tag{E.8}$$

We can then compute the following diagrams.

- $L_1$ and $L_2$

$$\text{(E.9)}$$

These two loops are special cases of $D_3$ and $D_4$. In the $\epsilon = d - 2$ expansion, we still adopt the propagator of $\sigma$ at $d = 2$. This gives

$$L_1[k, \omega] = -\frac{2}{c_2 N} D_3\left[1, -\frac{1}{2}; k, \omega\right] \approx \frac{8}{3\pi^2 N} \frac{1}{\epsilon}(k^2 + \omega^2)\mu^\epsilon, \tag{E.10}$$

$$L_2[k, \omega] = \frac{4W^2}{c_2^2 N} D_4[1, -1; k, \omega] \approx \frac{128W^2}{\pi N} \frac{1}{\epsilon}(\omega^2 - k^2)\mu^\epsilon. \tag{E.11}$$

- $L_3$ and $L_4$

$$\text{(E.12)}$$

These two loops are computed as

$$L_3[k, \omega] = -\int \frac{d^d p \, d\Omega}{(2\pi)^{d+1}} \frac{L_1[p, \Omega]}{(p^2 + \Omega^2)^2} \frac{1}{(k - p)^2 + (\omega - \Omega)^2} = -\frac{8}{3\pi^2 N} \frac{\mu^\epsilon}{\epsilon} D_3[1, 1; k, \omega]$$

$$= -\frac{1}{3\pi^2 N} \frac{\mu^\epsilon}{\epsilon}(k^2 + \omega_B^2)^{-1/2}, \tag{E.13}$$

$$L_4[k, \omega] = -\int \frac{d^d p \, d\Omega}{(2\pi)^{d+1}} \frac{L_2[p, \Omega]}{(p^2 + \Omega^2)^2} \frac{1}{(k - p)^2 + (\omega - \Omega)^2}$$

$$= -\frac{128W^2}{\pi N} \frac{\mu^\epsilon}{\epsilon} \left( D_3''[2, 1; k, \omega] - D_3'[2, 1; k, \omega] \right) = -\frac{8W^2}{\pi N} \frac{\mu^\epsilon}{\epsilon} \frac{k^2}{(k^2 + \omega^2)^{\frac{3}{2}}}. \tag{E.14}$$

- 3-vertex

  In order to compute more diagrams, we study the $\sigma\phi\phi$ vertices. Simple power counting tells that these diagrams are UV divergent. For the purpose of this paper, we only need to evaluate the divergent part. This means we are able to reconnect the external points to the equivalent internal ones such that the diagrams are made up of only subdiagrams listed in Eq. (E.2), namely $D_{1\sim 4}$. This trick of computing divergent part is illustrated comprehensively in [42].

– One loop corrections

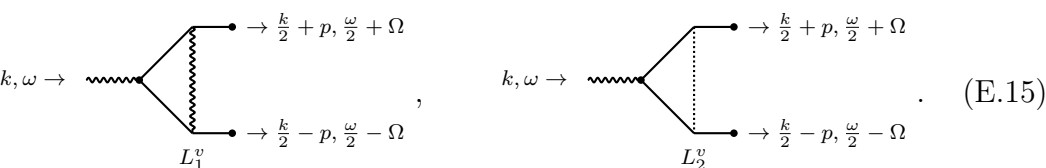

$$, \qquad . \tag{E.15}$$

These two loops are expressed as

$$L_1^v[k,p,\omega,\Omega] = \frac{-i2}{c_2 N^{\frac{3}{2}}} \int \frac{d^d q d\omega'}{(2\pi)^{d+1}} \frac{[(p-q)^2 + (\Omega-\omega')^2]^{\frac{1}{2}}}{[(\frac{k}{2}-q)^2 + (\frac{\omega}{2}-\omega')^2][(\frac{k}{2}+q)^2 + (\frac{\omega}{2}+\omega')^2]} \tag{E.16}$$

$$L_2^v[k,p,\omega,\Omega] = \frac{i4W^2}{c_2^2 N^{\frac{3}{2}}} \int \frac{d^d q}{(2\pi)^d} \frac{(p-q)^2}{[(\frac{k}{2}-q)^2 + (\frac{\omega}{2}-\Omega)^2][(\frac{k}{2}+q)^2 + (\frac{\omega}{2}+\Omega)^2]}. \tag{E.17}$$

Both integrals are divergent in the UV. After reconnecting the external and internal points, the resulting diagrams are shown below.

$$, \qquad . \tag{E.18}$$

They respectively have the same divergent parts as the two integrals above, i.e. $L_{1,2}^v \approx L_{1,2}^{v-div}$ as $\epsilon \to 0$. These two diagrams are given by

$$L_1^{v-div} = -i\frac{2}{c_2 N^{\frac{3}{2}}} \int \frac{d^d q d\omega'}{(2\pi)^{d+1}} \frac{1}{(\frac{k}{2}-p-q)^2 + (\frac{\omega}{2}-\Omega-\omega')^2} \frac{1}{[q^2 + (\omega')^2]^{\frac{1}{2}}}$$

$$= -i\frac{2}{c_2 N^{\frac{3}{2}}} D_3[1, \frac{1}{2}; \frac{k}{2}-p, \frac{\omega}{2}-\Omega] \approx i\frac{8}{\pi^2 N^{\frac{3}{2}}} \frac{\mu^\epsilon}{\epsilon}, \tag{E.19}$$

$$L_2^{v-div} = i\frac{4W^2}{c_2^2 N^{\frac{3}{2}}} \int \frac{d^d q}{(2\pi)^d} \frac{1}{(\frac{k}{2}-p-q)^2 + (\frac{\omega}{2}-\Omega)^2} = -i\frac{128W^2}{\pi N^{\frac{3}{2}}} \frac{\mu^\epsilon}{\epsilon}. \tag{E.20}$$

– Two loop corrections

$$, \qquad . \tag{E.21}$$

These two diagrams are expressed as

$$L_3^v[k,p,\omega,\Omega] = i\frac{4}{c_2^2 N^{\frac{3}{2}}} \int \frac{d^d q_1 d\omega_1}{(2\pi)^{d+1}} \frac{d^d q_2 d\omega_2}{(2\pi)^{d+1}} \frac{1}{(\frac{k}{2}-q_1)^2+(\frac{\omega}{2}-\omega_1)^2} \frac{1}{(\frac{k}{2}+q_1)^2+(\frac{\omega}{2}+\omega_1)^2}$$

$$\times \frac{[(\frac{k}{2}-q_2)^2+(\frac{\omega}{2}-\omega_2)^2]^{\frac{1}{2}}}{(q_2-q_1)^2+(\omega_2-\omega_1)^2} \frac{[(\frac{k}{2}+q_2)^2+(\frac{\omega}{2}+\omega_2)^2]^{\frac{1}{2}}}{(p-q_2)^2+(\Omega-\omega_2)^2}, \tag{E.22}$$

$$L_4^v[k,p,\omega,\Omega] = -i\frac{8W^2}{c_2^3 N^{\frac{3}{2}}} \int \frac{d^d q_1 d\omega_1}{(2\pi)^{d+1}} \frac{d^d q_2}{(2\pi)^{d}} \frac{1}{(\frac{k}{2}-q_1)^2+(\frac{\omega}{2}-\omega_1)^2} \frac{1}{(\frac{k}{2}+q_1)^2+(\frac{\omega}{2}+\omega_1)^2}$$

$$\times \frac{(\frac{k}{2}-q_2)^2}{(q_2-q_1)^2+(\frac{\omega}{2}-\omega_1)^2} \frac{[(\frac{k}{2}+q_2)^2+\omega^2]^{\frac{1}{2}}}{(p-q_2)^2+(\Omega-\frac{\omega}{2})^2}. \tag{E.23}$$

By reconnecting the external points to the equivalent internal points, we can get following diagrams with the same divergent parts, i.e. $L_{3,4}^v \approx L_{3,4}^{v-div}$ in the limit of $\epsilon \to 0$.

In the following, we found the divergent parts are both zero.

$$L_3^{v-div} = i\frac{4}{c_2^2 N^{\frac{3}{2}}} \int \frac{d^d q d\omega'}{(2\pi)^{d+1}} \frac{D_3[1,0,q,\omega']}{q^2+(\omega')^2} \frac{1}{(\frac{k}{2}-p-q)^2+(\frac{\omega}{2}-\Omega-\omega')^2} = 0, \tag{E.25}$$

$$L_4^{v-div} = -i\frac{8W^2}{c_2^3 N^{\frac{3}{2}}} \int \frac{d^d q d\omega'}{(2\pi)^{d+1}} D_4[1,-\frac{1}{2},q,\omega'] \frac{1}{q^2+(\omega')^2} \frac{1}{(\frac{k}{2}-p-q)^2+(\frac{\omega}{2}-\Omega-\omega')^2}$$

$$= i\frac{2W^2}{c_2^3 N^{\frac{3}{2}}} \frac{1}{(4\pi)^{\frac{d+1}{2}}} \frac{\Gamma[1-\frac{d}{2}]}{\Gamma[\frac{3}{2}]\Gamma^2[-\frac{1}{2}]} \int_0^1 dt dx dy \; t^{\frac{1}{2}}(1-t)^{-\frac{3}{2}} x^{-\frac{3}{2}}(1-xt)^{-\frac{d}{2}}$$

$$= i\frac{2W^2}{c_2^3 N^{\frac{3}{2}}} \frac{1}{(4\pi)^{\frac{d+1}{2}}} \frac{\Gamma[1-\frac{d}{2}]\Gamma[-\frac{1}{2}]}{\Gamma[\frac{3}{2}]\Gamma[-\frac{d}{2}]\Gamma[\frac{1}{2}]} {}_3F_2[\{-\frac{1}{2},-\frac{1}{2},-1\},\{-1,\frac{1}{2}\},1] = \text{finite}. \tag{E.26}$$

- $L_5$ and $L_6$

Now we are ready to compute the loop corrections of the $\sigma\sigma$ propagator. It is contributed by the following two diagrams.

Using the result of $L_{1,2}^{v-div}$, we can get

$$L_5[k,\omega] = i\sqrt{N} \int \frac{d^d p\, d\Omega}{(2\pi)^{d+1}} L_1^v[k,p,\omega,\Omega] \frac{1}{(\frac{k}{2}+p)^2 + (\frac{\omega}{2}+\Omega)^2} \frac{1}{(\frac{k}{2}-p)^2 + (\frac{\omega}{2}-\Omega)^2}$$

$$= -\frac{8}{\pi^2 N} \frac{\mu^\epsilon}{\epsilon} D_3[1,1;k,\omega] = -\frac{1}{\pi^2 N} \frac{\mu^\epsilon}{\epsilon} \frac{1}{\sqrt{k^2+\omega^2}}, \tag{E.28}$$

$$L_6[k,\omega] = i\sqrt{N} \int \frac{d^d p\, d\Omega}{(2\pi)^{d+1}} L_2^v[k,p,\omega,\Omega] \frac{1}{(\frac{k}{2}+p)^2 + (\frac{\omega}{2}+\Omega)^2} \frac{1}{(\frac{k}{2}-p)^2 + (\frac{\omega}{2}-\Omega)^2}$$

$$= \frac{128 W^2}{\pi N} \frac{\mu^\epsilon}{\epsilon} D_3[1,1;k,\omega] = \frac{16 W^2}{\pi N} \frac{\mu^\epsilon}{\epsilon} \frac{1}{\sqrt{k^2+\omega^2}}. \tag{E.29}$$

There would be a factor of 2 for each diagram coming from both left and right subdiagrams. However, it is cancelled by the symmetry factor $\frac{1}{2}$.

- $L_7$ and $L_8$

As shown before, there is no divergent subdiagram. Therefore, these two diagrams are finite.

## Appendix F: renormalization of composite operator $\phi^2$ at $4-\epsilon$

In this section, we compute the connected 2-point correlation function of the composite operator $\phi^2$. Up to the order of $\frac{1}{N}$, the bare correlation function is contributed by the following diagrams.

$$G_{\phi^2}^B = \qquad + \qquad + \qquad + \quad \mathcal{O}(1/N^2). \tag{F.1}$$

These diagrams are evaluated as following. Firstly, $L_{10}$ is given by

$$L_{10} = 2N D_3[1,1,p,\omega] = 2N c_2 [p^2 + \omega^2]^{\frac{d-3}{2}}, \tag{F.2}$$

where $c_2 = -\frac{2^{-(d+1)}(4\pi)^{\frac{2-d}{2}}}{\Gamma[\frac{d}{2}] \sin \frac{(d+1)\pi}{2}}$. At $d=4$, $c_2 = -\frac{1}{128\pi}$. Secondly, we can compute $L_{11}$ as

$$L_{11} = -\frac{8NW^2}{(4\pi)^2} \frac{\mu^{-\epsilon}}{\epsilon} D_3''[2,1,p,\omega] = \frac{W^2 N c_2}{8\pi^2} \frac{\mu^{-\epsilon}}{\epsilon} \left[ -[p^2+\omega^2]^{\frac{1}{2}} + \frac{\omega^2}{[p^2+\omega^2]^{\frac{1}{2}}} \right]. \tag{F.3}$$

At last, the diagram $L_{12}$ is expressed as

$$L_{12}[p,\omega] = 4NW^2 \int \frac{d^d k}{(2\pi)^d} \frac{d^d q}{(2\pi)^d} \frac{d\Omega}{2\pi} \frac{1}{k^2+\Omega^2} \frac{1}{q^2+\Omega^2} \frac{1}{(p-k)^2 + (\omega-\Omega)^2} \frac{1}{(p-q)^2 + (\omega-\Omega)^2} \tag{F.4}$$

It contains subdiagram

$$S[p, \omega; p - q, \omega - \Omega; q, \Omega] = \int \frac{d^d k}{(2\pi)^d} \frac{1}{k^2 + \Omega^2} \frac{1}{(p-k)^2 + (\omega - \Omega)^2} \approx \frac{\mu^{-\epsilon}}{8\pi^2 \epsilon}, \qquad (F.5)$$

which finally gives

$$L_{12}[p, \omega] = 4NW^2 \frac{\mu^{-\epsilon}}{8\pi^2 \epsilon} \int \frac{d^d q}{(2\pi)^d} \frac{d\Omega}{2\pi} \frac{1}{q^2 + \Omega^2} \frac{1}{(p-q)^2 + (\omega - \Omega)^2} = \frac{W^2 N}{2\pi^2} \frac{\mu^{-\epsilon}}{\epsilon} c_2 [p^2 + \omega^2]^{\frac{d-3}{2}} (F.6)$$

Consequently, the bare correlation function is

$$G_{\phi^2}^B = L_2 + L_3 + L_4 = 2N \left[ 1 + \left( \frac{3}{16\pi^2} + \frac{1}{16\pi^2} \frac{\omega^2}{p^2 + \omega^2} \right) \frac{W^2}{N} \frac{\mu^{-\epsilon}}{\epsilon} \right] c_2 [p^2 + \omega^2]^{\frac{1}{2}}. \qquad (F.7)$$

The divergence is cancelled by the counterterms which can fix the renormalization constant $Z_{\phi^2}$ to be

$$Z_{\phi^2} = 1 + \frac{3}{32\pi^2} \frac{W_R^2}{N} \frac{\mu^{-\epsilon}}{\epsilon} - \frac{1}{2} \delta_\tau = 1 + \frac{1}{16\pi^2} \frac{W_R^2}{N} \frac{\mu^{-\epsilon}}{\epsilon}, \qquad (F.8)$$

which is the same as Eq. (6.13).

## Appendix G: $\beta$ function of the randomness up to order $1/N^2$

The Gaussian distribution in Eq. (6.18) has higher moments of the disordered coupling as follows.

$$\overline{J_{k_1} J_{k_2}} = W^2 \delta^d(k_1 + k_2) + J_0^2 \delta^d(k_1) \delta^d(k_2), \qquad (G.1)$$

$$\overline{J_{k_1} J_{k_2} J_{k_3}} = W^2 J_0 \left[ \delta^d(k_1 + k_3) \delta^d(k_2) + \delta^d(k_2 + k_3) \delta^d(k_1) + \delta^d(k_1 + k_2) \delta^d(k_3) \right]$$
$$+ J_0^3 \delta^d(k_1) \delta^d(k_2) \delta^d(k_3), \qquad (G.2)$$

$$\overline{J_{k_1} J_{k_2} J_{k_3} J_{k_4}} = W^4 \left[ \delta^d(k_1 + k_2) \delta^2(k_3 + k_4) + \delta^d(k_1 + k_3) \delta^2(k_2 + k_4) + \delta^d(k_1 + k_4) \delta^2(k_2 + k_3) \right]$$
$$+ W^2 J_0^2 \left[ \delta^d(k_1 + k_2) \delta^d(k_3) \delta^d(k_4) + \delta^d(k_1 + k_3) \delta^d(k_2) \delta^d(k_4) + \delta^d(k_1 + k_4) \delta^d(k_2) \delta^d(k_3) \right.$$
$$+ \left. \delta^d(k_3 + k_4) \delta^d(k_1) \delta^d(k_2) + \delta^d(k_2 + k_4) \delta^d(k_1) \delta^d(k_3) + \delta^d(k_2 + k_3) \delta^d(k_1) \delta^d(k_4) \right]$$
$$+ J_0^4 \delta^d(k_1) \delta^d(k_2) \delta^d(k_3) \delta^d(k_4), \qquad (G.3)$$

$$\overline{J_{k_1} J_{k_2} J_{k_3} J_{k_4} J_{k_5}} = W^4 J_0 \sum_{i=1}^{5} \sum_{j_a \neq i} \delta^d(k_i) [\delta^d(k_{j_1} + k_{j_2}) \delta^d(k_{j_3} + k_{j_4}) + \delta^d(k_{j_1} + k_{j_3}) \delta^d(k_{j_2} + k_{j_4})$$
$$+ \delta^d(k_{j_1} + k_{j_4}) \delta^d(k_{j_2} + k_{j_3})] + W^4 J_0^3 \left[ \delta^d(k_1 + k_2) \delta^d(k_3) \delta^d(k_4) \delta^d(k_5) \right.$$
$$+ \delta^d(k_1 + k_3) \delta^d(k_2) \delta^d(k_4) \delta^d(k_5) + \delta^d(k_1 + k_4) \delta^d(k_2) \delta^d(k_3) \delta^d(k_5)$$
$$+ \delta^d(k_1 + k_5) \delta^d(k_2) \delta^d(k_3) \delta^d(k_4) + \delta^d(k_2 + k_3) \delta^d(k_1) \delta^d(k_4) \delta^d(k_5)$$
$$+ \delta^d(k_2 + k_4) \delta^d(k_1) \delta^d(k_3) \delta^d(k_5) + \delta^d(k_2 + k_5) \delta^d(k_1) \delta^d(k_3) \delta^d(k_4)$$
$$+ \delta^d(k_3 + k_4) \delta^d(k_1) \delta^d(k_2) \delta^d(k_5) + \delta^d(k_3 + k_5) \delta^d(k_1) \delta^d(k_2) \delta^d(k_4)$$
$$+ \left. \delta^d(k_4 + k_5) \delta^d(k_1) \delta^d(k_2) \delta^d(k_3) \right] + J_0^5 \delta^d(k_1) \delta^d(k_2) \delta^d(k_3) \delta^d(k_4) \delta^d(k_5). \qquad (G.4)$$

Up to the order of $1/N^2$, the disorder coupling of the singlet operator is

$$\mathcal{J}_{k,p} = J_{k-p} - \frac{1}{\sqrt{N}} \int_{\Lambda e^{-\ell}}^{\Lambda} \frac{d^d q}{(2\pi)^d} J_{k-q} J_{q-p} G^>(q,\omega) + \frac{1}{N} \int_{\Lambda e^{-\ell}}^{\Lambda} \frac{d^d q_1 d^d q_2}{(2\pi)^{2d}} J_{k-q_1} J_{q_1-q_2} J_{q_2-p} G^>(q_1,\omega) G^>(q_2,\omega)$$

$$- \frac{1}{N^{\frac{3}{2}}} \int_{\Lambda e^{-\ell}}^{\Lambda} \frac{d^d q_1 d^d q_2 d^d q_3}{(2\pi)^{3d}} J_{k-q_1} J_{q_1-q_2} J_{q_2-q_3} J_{q_3-p} G^>(q_1,\omega) G^>(q_2,\omega) G^>(q_3,\omega)$$

$$+ \frac{1}{N^2} \int_{\Lambda e^{-\ell}}^{\Lambda} \frac{d^d q_1 d^d q_2 d^d q_3 d^d q_4}{(2\pi)^{4d}} J_{k-q_1} J_{q_1-q_2} J_{q_2-q_3} J_{q_3-q_4} J_{q_4-p} G^>(q_1,\omega) G^>(q_2,\omega) G^>(q_3,\omega) G^>(q_4,\omega) \quad \text{(G.5)}$$

In order to get the distribution of $\mathcal{J}_{k,p}$, we can compute the mean and variance as

$$\overline{\mathcal{J}_{k,p}} = \overline{J_{k-p}} - \frac{1}{\sqrt{N}} \int_{\Lambda e^{-\ell}}^{\Lambda} \frac{d^d q}{(2\pi)^d} \overline{J_{k-q} J_{q-p}} G^>(q,\omega) + \frac{1}{N} \int_{\Lambda e^{-\ell}}^{\Lambda} \frac{d^d q_1 d^d q_2}{(2\pi)^{2d}} \overline{J_{k-q_1} J_{q_1-q_2} J_{q_2-p}} G^>(q_1,\omega) G^>(q_2,\omega)$$

$$- \frac{1}{N^{\frac{3}{2}}} \int_{\Lambda e^{-\ell}}^{\Lambda} \frac{d^d q_1 d^d q_2 d^d q_3}{(2\pi)^{3d}} \overline{J_{k-q_1} J_{q_1-q_2} J_{q_2-q_3} J_{q_3-p}} G^>(q_1,\omega) G^>(q_2,\omega) G^>(q_3,\omega)$$

$$+ \frac{1}{N^2} \int_{\Lambda e^{-\ell}}^{\Lambda} \frac{d^d q_1 d^d q_2 d^d q_3 d^d q_4}{(2\pi)^{4d}} \overline{J_{k-q_1} J_{q_1-q_2} J_{q_2-q_3} J_{q_3-q_4} J_{q_4-p}} G^>(q_1,\omega) G^>(q_2,\omega) G^>(q_3,\omega) G^>(q_4,\omega)$$

$$+ \mathcal{O}(1/N^{\frac{5}{2}}) \quad \text{(G.6)}$$

and

$$\overline{\mathcal{J}_{k_1,p_1} \mathcal{J}_{k_2,p_2}} = \overline{J_{k_1-p_1} J_{k_2-p_2}} - \frac{1}{\sqrt{N}} \int_{\Lambda e^{-\ell}}^{\Lambda} \frac{d^d q}{(2\pi)^d} \overline{J_{k_1-p_1} J_{k_2-q} J_{q-p_2}} G^>(q,\omega)$$

$$- \frac{1}{\sqrt{N}} \int_{\Lambda e^{-\ell}}^{\Lambda} \frac{d^d q}{(2\pi)^d} \overline{J_{k_2-p_2} J_{k_1-q} J_{q-p_1}} G^>(q,\omega)$$

$$+ \frac{1}{N} \int_{\Lambda e^{-\ell}}^{\Lambda} \frac{d^d q_1 d^d q_2}{(2\pi)^{2d}} \overline{J_{k_1-q_1} J_{q_1-p_1} J_{k_2-q_2} J_{q_2-p_2}} G^>(q_1,\omega) G^>(q_2,\omega)$$

$$+ \frac{2}{N} \int_{\Lambda e^{-\ell}}^{\Lambda} \frac{d^d q_1 d^d q_2}{(2\pi)^{2d}} \overline{J_{k_1-p_1} J_{k_2-q_1} J_{q_1-q_2} J_{q_2-p_2}} G^>(q_1,\omega) G^>(q_2,\omega)$$

$$- \frac{2}{N^{\frac{3}{2}}} \int_{\Lambda e^{-\ell}}^{\Lambda} \frac{d^d q_1 d^d q_2 d^d q_3}{(2\pi)^{3d}} \overline{J_{k_1-q_3} J_{q_3-p_1} J_{k_2-q_1} J_{q_1-q_2} J_{q_2-p_2}} G^>(q_1,\omega) G^>(q_2,\omega) G^>(q_3,\omega)$$

$$- \frac{2}{N^{\frac{3}{2}}} \int_{\Lambda e^{-\ell}}^{\Lambda} \frac{d^d q_1 d^d q_2 d^d q_3}{(2\pi)^{3d}} \overline{J_{k_1-p_1} J_{k_2-q_1} J_{q_1-q_2} J_{q_2-q_3} J_{q_3-p_2}} G^>(q_1,\omega) G^>(q_2,\omega) G^>(q_3,\omega)$$

$$+ \frac{2}{N^2} \int_{\Lambda e^{-\ell}}^{\Lambda} \frac{d^d q_1 d^d q_2 d^d q_3 d^d q_4}{(2\pi)^{4d}} \overline{J_{k_1-p_1} J_{k_2-q_1} J_{q_1-q_2} J_{q_2-q_3} J_{q_3-q_4} J_{q_4-p_2}}$$

$$\times G^>(q_1,\omega) G^>(q_2,\omega) G^>(q_3,\omega) G^>(q_4,\omega) + \mathcal{O}(1/N^{\frac{5}{2}}). \quad \text{(G.7)}$$

For the external momenta in the long wavelength limit, to the first order of $\ell$, we can get

$$\overline{\mathcal{J}_{k,p}} = \left[ J_0 - \frac{W^2}{\sqrt{N}} \eta_1 + \frac{W^2 J_0}{N} \eta_2 - \frac{W^2 J_0^2}{N^{\frac{3}{2}}} \eta_3 + \frac{W^2 J_0^3}{N^2} \eta_4 \right] \delta^d(k-p), \quad \text{(G.8)}$$

$$\overline{\mathcal{J}_{k_1,p_1} \mathcal{J}_{k_2,p_2}} - \overline{\mathcal{J}_x}\,\overline{\mathcal{J}_{x'}} = \left[ W^2 + \frac{2W^4}{N} \eta_2 - \frac{16 W^4 J_0}{N^{\frac{3}{2}}} \eta_3 + \frac{6 W^4 J_0^2}{N^2} \eta_4 \right] \delta^d(k_1 + k_2 - p_1 - p_2), \text{(G.9)}$$

where

$$\eta_m = \int_{\Lambda e^{-\ell}}^{\Lambda} \frac{d^d q}{(2\pi)^d} [G^>(q,0)]^m = \frac{\Lambda^{4-2m}\ell}{8\pi^2}. \qquad (G.10)$$

Then, under rescaling of the spacetime, the disordered coupling requires a redefinition as $\mathcal{J}_x \to \mathcal{J}_{x e^{\ell}} e^{-2\ell}$. As a result, the mean value and the variance of the Gaussian distribution at longer length scale becomes $\mathcal{J}_0 = \frac{J_0}{\Lambda^2} \to \frac{J_0}{\Lambda^2} + \ell\beta_J$ and $W^2 \to W^2 + \ell\beta_{W^2}$, where

$$\beta_J = \frac{d\mathcal{J}_0}{d\ell} = 2\mathcal{J}_0 - \frac{W^2}{8\pi^2\sqrt{N}} + \frac{W^2\mathcal{J}_0}{8\pi^2 N} - \frac{W^2\mathcal{J}_0^2}{8\pi^2 N^{\frac{3}{2}}} + \frac{W^2\mathcal{J}_0^3}{8\pi^2 N^2}, \qquad (G.11)$$

$$\beta_{W^2} = \frac{dW^2}{d\ell} = \epsilon W^2 + \frac{W^4}{4\pi^2 N} - \frac{2W^4\mathcal{J}_0}{\pi^2 N^{\frac{3}{2}}} + \frac{3W^4\mathcal{J}_0^2}{4\pi^2 N^2}. \qquad (G.12)$$

including higher order terms up to the order of $1/N^2$ correcting Eq. (6.31) and Eq. (6.33). From these two $\beta$ functions, we still cannot find any physical stable fixed point. If we keep one more term in each of them compared with Eq. (6.31) and Eq. (6.33), we can find two solutions satisfying $\beta = 0$ at

$$\frac{\mathcal{J}_0^*}{\sqrt{N}} = -\frac{\epsilon}{4} + \mathcal{O}(\epsilon^2) \quad , \frac{(W^2)^*}{N} = -4\pi^2\epsilon + \mathcal{O}(\epsilon^2) \qquad (G.13)$$

and

$$\frac{\mathcal{J}_0^*}{\sqrt{N}} = \frac{1}{8} + \mathcal{O}(\epsilon) \quad , \frac{(W^2)^*}{N} = \frac{16\pi^2}{7} + \mathcal{O}(\epsilon) \qquad (G.14)$$

The first one is unphysical because it locates at negative $W^2$. While the second one is at positive $W^2$. It is at couplings of order one as $\epsilon \to 0$. This makes all the higher ordered corrections of the same order, which signals a breakdown of the perturbation. Thus, the second fixed point found within this framework is not reliable. Generally speaking, up to the order of $1/N^2$, there is no physical stable fixed point found by the perturbation method.