# Peer review of "Quenched random mass disorder in the large N theory of vector bosons"

_SciPost Physics_

## Round 1 · Referee Report · Ilya Esterlis (Referee 1) · 2022-10-24

Strengths

1. Very clearly written
2. Novel results on an important and outstanding problem

Report

In "Quenched random mass disorder in the large N theory of vector bosons," the author analyzes the problem of the quantum O(N) model with random mass disorder, using a combination of large-N and epsilon-expansion techniques. The problem of quantum systems with disorder is notoriously challenging, primarily due to the long-range imaginary-time correlations induced by the disordered couplings. Thus, new, controlled approaches to the problem are both welcome and of significant importance. In particular, approaches that avoid the "replica trick" provide a useful complement to the existing literature.

The primarily results of the paper are (1) An exact solution of the problem at N=infinity. An especially interesting point is the author's discovery of the necessity of an "intrinsic scale" in the problem for space dimension 2<d<3. This leads to a scale invariant theory in a reduced number of dimensions, akin to the phenomenon of "dimensional reduction" in the random-field Ising model. (2) The calculation of leading 1/N corrections. A highlight of this is the author's demonstration that there exists a disordered, interacting fixed point that is stable to 1/N corrections for d=2+epsilon dimensions. One may therefore hope that the results of the current paper could even be extrapolated to the physically relevant case of d=3 (epsilon=1).

I find the exposition to be especially clear and pedagogical. The fact that the author computes things using both the "condensed matter/statistical mechanics" and "high-energy" versions of RG is particularly helpful.

I have a few minor questions/comments for the author:

1. The author makes several references to the notion of "generalized free fields." While this is discussed in Ref. 30, I think the paper would benefit from a very brief reminder/explanation of this idea. My pedestrian understanding of generalized free fields is something like mean-field theory (which indeed solves the O(N) model at N=infinity) -- that is, Gaussian theories with non-standard Green's functions, determined by solving an appropriate self-consistency equation (but perhaps my understanding is incorrect).

2. Something that comes up often in RG studies of quantum systems with quenched disorder are complex critical exponents with spiraling RG flows (e.g., Ref. 3 of the current paper). It appears such behavior is absent in these new calculations. Does the author have an understanding of this? Is it related to the instability of the replicated theory? I believe complex exponents have been observed in holographic calculations which do not rely on the replica trick; e.g., Hartnoll, Sean A., David M. Ramirez, and Jorge E. Santos. "Thermal conductivity at a disordered quantum critical point." JHEP 04 (2016).

3. In Sec. IVB, when discussing the dimensional reduction, there appears to be something singular happening when d-> 3 (the dimension of the associated free CFT diverges). Perhaps I missed it, but is there an explanation or understanding of this singularity.

I believe this is a high-quality paper that offers a straight-forward and welcome new approach to the problem of quenched disorder in quantum systems, and I recommend it for publication.

  • validity: top
  • significance: top
  • originality: top
  • clarity: top
  • formatting: perfect
  • grammar: excellent

Author:  Han Ma  on 2022-11-21  [id 3056]

(in reply to Report 1 by Ilya Esterlis on 2022-10-24)

We thank the referee for reviewing our manuscript and for taking the time to make detailed comments and questions. We are glad that the referee recommends our work for publication. Here is our response to the referee’s comments and questions.

1) The generalized free field is a scaling operator in a CFT that has a different scaling dimension from that of the free field. If it has scaling dimension $\Delta$, then $\Delta \neq \frac{d-1}{2}$ in d spatial dimensions. It has non-zero two point function with form $\frac{1}{r^{2\Delta}}$ but its higher point (connected) functions are all zero. A simple theory for a generalized free field $\Phi$ is a Gaussian theory with action $S=\int d^{d+1} r~ \Phi (-\nabla^2)^{\Delta-\frac{d+1}{2}}\Phi$. We have added this explanation of the terminology "generalized free field" as a footnote on p.g.12.

2)Yes, the complex critical exponent and spiral RG flow are absent in this 1/N expansion calculation. In the double epsilon expansion in e.g. Ref.3, the perturbation is done around the free Gaussian fixed point, where there is accidental degeneracy of the operators. More concretely, the operator $\phi^4(x,\tau)$ and $\phi^2(x,\tau)\phi^2(x,\tau') $ are both relevant and have the same scaling dimension. Using the replica trick or not, the latter operator would always contribute to the disorder averaged correlation functions. These two operators are mixed along the RG acquiring complex scaling dimensions and induce the spiral flow. However, there is no such an accidental degeneracy at the infinite N fixed point of the O(N) vector model. So perturbation around this fixed point will not result in any complex scaling dimension.

3)The derivation in Sec.IV is for the system below 3 spatial dimensions. The divergence in the dimension reduction at $d\rightarrow 3$ is traced back to $G_\sigma \rightarrow 0$ in Eq.(4.1) at $d\rightarrow 3$. In fact, as $d\rightarrow 3$, we should correct the action by the term $\sigma^2$ which is exactly marginal in the infinite N limit. In this way, the propagator of the $\sigma$ field becomes a constant and the effective action in Eq.(4.15) can be written in terms of the bosonic $\sigma$ field only. After the integration of field $\sigma$ at nonzero frequency, we get an effective action at zero frequency which can be directly identified as a scale invariant theory in 3 dimensions. This is consistent with our conclusion that the O(N) model in a d+1 spacetime dimensional system is scale invariant in the 3 dimensional subsystem. We have added a brief analysis on the d=3 system at the end of IV B on p.g. 17.

---

## Round 1 · Referee Report · Anonymous (Referee 2) · 2022-11-1

Strengths

1. Valid
2. Clear presentation
3. New approach to problem

Weaknesses

1. Results are somewhat incremental

Report

This paper approaches the problem of the O(N) model with random mass disorder at large N. This theory has been studied before by other works.

The novelty of this paper is that a somewhat different approach is used, where, instead of working with the replicated disorder-averaged action in the limit of n (number of replicas) going to 0, the author works with the action expanded perturbatively in the disordered term, with disorder-averaging then performed directly order by order. Because of the large N limit, the series can be truncated at a finite order in order to obtain correction effects up to some desired power of 1/N.

The the author computes corrections to scaling dimensions using standard Wilsonian field-theoretic RG techniques. The results of Ref. [6] are then extended to d = 2+\epsilon instead of just d = 2, extrapolating these results gives the possibilty of a novel fixed point at d = 3.

The paper is correct as far as I can see, and the presentation is sufficiently clear. I recommend publication in SciPost.

  • validity: top
  • significance: good
  • originality: ok
  • clarity: top
  • formatting: excellent
  • grammar: excellent

Author:  Han Ma  on 2022-11-21  [id 3055]

(in reply to Report 2 on 2022-11-01)

We thank the referee for reviewing this manuscript and thank her/him for agreeing that our manuscript is suitable for publication in SciPost. While we recognize the referee’s assessment is generally positive, she/he somehow thinks our result is incremental. Regarding this critique, we would like to list our new results which were unknown before. 1)As addressed by the referee, we push the study of the disordered fixed point of the O(N) model to 2+epsilon spatial dimensions by double expansion of epsilon and 1/N. 2)At infinite N, we can actually get the disordered IR physics at arbitrary dimensions. The disorder strength plays a role of intrinsic length scale and the whole system has scale invariance in a 3 dimensional subsystem. After extensive studies on the classical random field Ising model, this is the first example of dimension reduction in the system with quantum quenched disorder. 3) Besides, around 4 spatial dimensions, we have obtained the fixed point structure and an unstable fixed point is found above 4 dimensions. This is the first time we are able to study a disordered system above the upper critical dimension where the replica method is invalid.

---

## Round 2 · List of Changes

1.We have added the explanation of "generalized free field" as a footnote on p.g.12.
2. We have added a brief analysis of dimension reduction for the d=3 system at the end of IV B on p.g. 17.

---

## Editorial Decision

editorial_decision: